# Multilocus sequence typing and antibiotic resistance of *Aeromonas* isolated from freshwater fish in Hebei Province

Zixiao Yu[1‡], Yunseok Oh[2‡], Songmi Kim[3], Kyudong Han[2,3,4,5,6], Kornsorn Srikulnath[7], Qingyang Li[8], Ji-Seok Jang[9]*, Ho-Seong Lee[1]*

1 Department of Exercise and Medical Science, Graduate School, Dankook University, Cheonan, Republic of Korea, 2 Department of Microbiology, College of Science & Technology, Dankook University, Cheonan, Republic of Korea, 3 Smart Animal Bio institute, Dankook University, Cheonan, Republic of Korea, 4 Center for Bio Medical Engineering Core Facility, Dankook University, Cheonan, Republic of Korea, 5 Department of Bioconvergence Engineering, Dankook University, Jukjeon, Republic of Korea, 6 HuNBiome Co., Ltd., R&D Center, Geumcheon-gu, Seoul, Korea, 7 Animal Genomics and Bioresource Research Unit (AGB Research Unit), Faculty of Science, Kasetsart University, Bangkok, Thailand, 8 College of Life Sciences, Hebei Normal University, Shijiazhuang, China, 9 Department of Sport Management, College of Sport Science, Dankook University, Cheonan, Republic of Korea

‡ ZY and YO are contributed equally to this work and share first authorship.
* hoseh28@dankook.ac.kr (HSL); longlivesports@dankook.ac.kr (JSJ)

**Data Availability Statement:** Anyone can access the data for free. Please send queries to 'Molecular Bacteriology Laboratory of Hebei Normal University' (contact via E.mail: hbnumicrobe@163.

## Abstract

*Aeromonas* spp. are the opportunistic pathogens that infect both aquatic and terrestrial homeotherms. They were commonly present in aquatic environments, including effluent, tap water, marine, river, and lake, where they are often isolated from aquatic animals, including fish, molluscs, and crustaceans. The *Aeromonas* infections can cause sepsis, ulcer, and other symptoms, resulting in the death of massive aquatic animals. Therefore, the prevention and control of *Aeromonas* is of great significance for the healthy development of aquaculture. In this study, we used modern molecular methods to enhance disease control of *Aeromonas* isolates from freshwater fish in Hebei Province. A total of 130 *Aeromonas* spp. isolates were isolated from freshwater fish farms in Hengshui, Handan, and Shijiazhuang and all 130 *Aeromonas* spp. isolates were sequenced for species identification. Of the 130 *Aeromonas* spp. isolates, 104 isolates were successfully sequenced, and BLAST analysis showed that *Aeromonas veronii* was predominant in freshwater fish farms in Hebei Province. In addition, 26 antibiotic resistance profiles were obtained from 102 fully cultured isolates among the 104 *Aeromonas* spp. isolates whose species was primarily identified, and 44 multidrug-resistant bacteria among the 102 isolates were identified using an antibiotic susceptibility test. Using the Multilocus Sequence Typing (MLST) method, 33 out of 44 multidrug-resistant isolates with 14 non-*Aeromonas* reference strains were selected for phylogenetic and MLST analysis, and all 33 multidrug-resistant isolates were *A. veronii*. A total of 30 new Sequence Types (STs) were obtained by comparing concatenated sequences (*gyrB-groL-gltA-metG-ppsA-recA*) on PubMLST website. Furthermore, recombination event analysis detected using RDP5 and ClonalFrameML software 42 and 49 recombination events, respectively, and 22 recombination events were validated by four or more algorithms. Since mutation and recombination events increase clonal diversity and single

com) Multilocus sequence typing (MLST) profile data are available from the PubMLST database (https://pubmlst.org/). The data ID is 1038 - 1070 Input the number 1038 - 1070 into the search box to find the MLST profile data in this experiment (https://pubmlst.org/bigsdb?db=pubmlst_aeromonas_isolates&l=1&page=query).

**Funding:** This research was supported by Basic Science Research Program through the National Research Foundation of Korea (NRF) funded by the Ministry of Education (NRF-RS-2023-00275307) and Basic Science Research Capacity Enhancement Project through Korea Basic Science Institute (National research Facilities and Equipment Center) grant funded by the Ministry of Education (Grant No. 2019R1A6C1010033). The funder [NRF-RS-2023-00275307/Grant No. 2019R1A6C1010033] contributed to the decision to publish, manuscript preparation, and recruiting staff.

**Competing interests:** This research was supported by Basic Science Research Program through the National Research Foundation of Korea (NRF) funded by the Ministry of Education (NRF-RS-2023-00275307) and Basic Science Research Capacity Enhancement Project through Korea Basic Science Institute (National research Facilities and Equipment Center) grant funded by the Ministry of Education (Grant No. 2019R1A6C1010033).The funder [NRF-RS-2023-00275307/Grant No. 2019R1A6C1010033] contributed to the decision to publish, manuscript preparation, and recruiting staff.

housekeeping gene sequence alignments are limited for identifying species, we propose the use of multiple concatenated sequence loci to increase discriminatory power. In addition, we propose that the MLST method is an appropriate technique to study and develop the resistance mechanisms of multidrug-resistant *Aeromonas* and to identify *Aeromonas* systematically in complex samples obtained from the environment.

## Introduction

*Aeromonas* spp. are known as opportunistic pathogens capable of causing various diseases in aquatic animals, livestock, and humans and are widely distributed in ecosystems encompassing rivers, lakes, oceans, and urban domestic water [1, 2]. *Aeromonas* spp. possess various virulence factors such as degradative enzymes, surface polysaccharides (capsule, lipopolysaccharide, and glucan), S-layers, iron-binding systems, exotoxins, secretion systems, fimbriae, other nonfilamentous adhesins, and flagella [3, 4]. Various virulence factors of *Aeromonas* spp. cause immunocompromised organisms through resistance to host defense mechanisms, leading to diseases such as septicemia, ulcers, and visceral erosion and directly damaging host cells, resulting in *Aeromonas* infection and cell death [5, 6]. In addition, antibacterial and antioxidant molecules in the fish body provide self-defense against invading *Aeromonas* spp. and virulence factors, but the end product of this is reactive oxygen species (ROS), which can cause host cell damage and stress [7]. Aquatic animals such as fish, shrimp, and amphibians are particularly susceptible to *Aeromonas* infection due to *Aeromonas* spp. being the most widespread species in the aquatic environment, which can result in high mortality [8, 9]. Over the past few years, fish diseases caused by *Aeromonas* spp. have increased, including *A. hydrophila*, *A. sobria*, *A. veronii* and *A. caviae* [10]. In particular, *A. hydrophila* and *A. veronii* cause hemorrhagic diseases in farmed fish, widely infect aquatic animals, and quickly become the epicenter of epidemic outbreaks, resulting in significant economic losses in aquaculture [11–14].

Multiple drug resistance (MDR) has been increased all over the world that is considered a public health threat [15–17]. Several recent investigations reported the emergence of multidrug-resistant bacterial pathogens from different origins that increase the necessity of the proper use of antibiotics [18, 19]. Besides, the routine application of the antimicrobial susceptibility testing to detect the antibiotic of choice as well as the screening of the emerging MDR strains [20, 21]. The occurrence of various diseases caused by *Aeromonas* spp. have prompted the use of antimicrobial drugs, including aminoglycosides, carbapenems, broad-spectrum cephalosporins, chloramphenicol, quinolones, tetracyclines and other antibiotics [22]. However, bacterial multidrug resistance has become an important research focus due to the regular use of these drugs and other factors contributing to the development of resistance. Some *Aeromonas* spp. exhibit varying levels of antibiotic resistance, depending primarily on species, distribution, and other factors. In particular, resistance to ampicillin and penicillin is commonly observed in *Aeromonas* spp. [23, 24]. In fact, some *Aeromonas* spp. harbor genes encoding $\beta$-lactamases, which confer resistance to $\beta$-lactam antibiotics such as penicillin, carbapenems, and monocyclic-$\beta$-lactams [25]. The research at this stage can guide the rational use of antibiotics in various industries to reduce the probability of resistance development. Therefore, in the context of understanding the molecular mechanisms, it becomes imperative to identify antimicrobial resistance genes associated with phenotypic resistance and analyze the underlying causes of drug resistance in strains.

Accurate classification of *Aeromonas* spp. is essential to determine whether there is a correlation between the occurrence of *Aeromonas* infections and the pathogenic mechanisms associated with their host and transmission routes. To elucidate the complex taxonomic relationships between *Aeromonas* strains, phylogenetic analysis based on the *16S rRNA* gene was first applied in large-scale identification of *Aeromonas*. However, the high degree of conservation of the *16S rRNA* gene limited the classification of some species. It has been suggested that relying solely on a single gene to identify the *Aeromonas* genus may lead unreliable outcomes due to its insufficient phylogenetic power at the species level and limited discriminatory power in some genera [26]. In 1995, it was discovered by Yamamoto that the *gyrB* gene may be a more appropriate phylogenetic marker for bacterial taxonomy [27].

Multilocus Sequence Typing (MLST) is a widely used bacterial typing method, first proposed in 1998 [28]. MLST approach is based on the sequence polymorphism within internal segments of housekeeping genes and typically involves the PCR amplification and sequencing of 6–11 housekeeping genes. By comparing the resulting sequences and assigning allele numbers, isolates can be assigned sequence types based on unique combinations of allele variants. MLST for classification is characterized by simplicity, high reproducibility, and discriminatory power and can more accurately determine the branching order in the evolution of species [29, 30]. Thus, compared to other typing methods, such as DNA-DNA hybridization, MLST provides a more reliable means of elucidating phylogenetic relationships among taxa [31].

In this study, we isolated 130 *Aeromonas* spp. from freshwater fish in Hebei Province, China, and determined the predominant species among them. To evaluate the drug resistance profiles of the identified *Aeromonas* isolates, we performed antibiotic susceptibility testing based on the current operational guidelines of the Clinical and Laboratory Standards Institute (CLSI) [32, 33]. Additionally, we investigated how genotype can influence phenotype by confirming drug resistance genes associated with specific resistance phenotypes. We also applied the Multilocus Sequence Typing (MLST) method to perform phylogenetic typing of the multidrug-resistant *Aeromonas* discovered in this study and identify new sequence types (STs). Therefore, this study aims to investigate the antibiotic resistance of *Aeromonas* from aquatic organisms in Hebei Province, China, and verify whether MLST technology is suitable for classifying multidrug-resistant *Aeromonas* isolated from the environment.

## Materials and methods

### Sample collection

All the experiments in this study were conducted with the approval of the Animal Ethics Committee of Hebei Normal University (Protocol Number: 198012). *Aeromonas* often causes disease outbreaks in freshwater cultured fish, resulting in *Aeromonas* sepsis and ulcerative infection. Some fish have bleeding spots on the body surface and fins [34]. Infected fish eat less and swim on the surface of the pond [35]. Select the fish with insufficient vitality in the pond and pick them up with a small fishing net for body surface sampling and fish were released following sampling. Samples were collected from four common freshwater fish, including grass carp, carp, crucian carp, and koi carp, obtained from freshwater fish farms in Hebei Province (Hengshui, Handan, and Shijiazhuang), China, from April 2020 to August 2021. Use the sterile cotton swab to sample the bleeding points on the body surface or the ulcers around the fins.

### Bacterial isolation and extraction of genomic DNA

The samples were aseptically cultured on *Aeromonas* isolation medium RS medium (Hopebio, Qingdao, China) using the streak plate method at 30°C for 18–24 hours, after that, the sterile inoculation loops were used to pick up the single yellow colonies and streak them on clean RS

medium to obtain pure cultures. The purified strains were preserved in 30% glycerol and stored in an ultra-low temperature refrigerator at -80˚C for long-term preservation. Genomic DNAs from the pure isolated strains were extracted using the boiling method [36]. Pure isolated strains were inoculated into 1mL of brain heart infusion broth (BHIB) (TOPBIO, Yantai, China) and cultured overnight in a shaker at 180 rpm, at 30˚C. The culture was then centrifuged at 10,000 rpm for 2 minutes to collect the bacterial pellet. Subsequently, 1 mL of sterile water was added to resuspend the pellet, followed by another round of centrifugation at 10,000 rpm for 1 min to discard the supernatant. This wash step was repeated 2 times. Bacteria were resuspended by adding 50 uL of sterile water and the suspension was heated in a boiling water bath for 15 min. When the boiling reaction was completed, the boiling lysed samples were in an ice-water bath for 10 minutes and then centrifuged at 20,000 rpm for 2 minutes. After the above steps, the supernatant was collected for genomic DNA and stored in a medical refrigerator at -20˚C.

## Sequencing of *16S rRNA* and *gyrB* genes

The genomic DNAs were amplified using primer pairs suitable to each target gene for identification, relying on the sequences of the universal bacterial *16S rRNA* and *gyrB* genes. The primer sequences are as follows: universal bacterial *16S rRNA*_F forward primer (5′-AGA GTTTGATCCTGGCTCAG-3′), universal bacterial *16S rRNA*_R reverse primer (5′-TACGA CTTAACCCCAATCG-3′), *gyrB*_F forward primer (5′-GGGGTCTACTGCTTCACCAA-3′), and *gyrB*_R reverse primer (5′-CTTGTCCGGGTTGTACTCGT-3′) [37, 38]. Polymerase chain reaction (PCR) was employed for gene amplification, using a 20ul reaction mixture: 1ul of genomic DNA template, 10 uL of 2X SuperMix (TransGen Biotech, Beijing, China), 1 uL each of forward and reverse primers, and 7 uL of ddH2O, resulting in a total volume of 20 uL. Following amplification, 5 uL of the PCR product from each sample was subjected to 1% agarose gel electrophoresis at 120V for 30 min. Samples exhibiting specific bands were documented using a gel imaging system (Bio-Rad, CA, USA) and then sent to GENEWIZ Co., Ltd. for sequencing (GENEWIZ, Suzhou, China).

## Sequence analysis and construction of the phylogenetic analysis

The sequencing results were compared with BLAST on the NCBI website (https://blast.ncbi. nlm.nih.gov/Blast.cgi), and a similarity ≥ 97% were considered reliable, and the result with the highest score was recorded [39]. The *16S rRNA* and *gyrB* sequences were combined and collated separately using the sequence merge function of the TBtools software [40]. Genetic diversity parameters such as G+C content, number of polymorphic loci, Tajima's D, nucleotide diversity ($\pi$), number of segregating sites ($\theta$), number of parsimony informative sites, synonymous substitution rate ($K_s$), and non-synonymous substitution rate ($K_a$) were calculated using DnaSP v6.12.03 [41]. The sequences were imported into the MEGA11 software and the nucleotide sequence and amino acid sequence comparison were performed using the MUSCLE algorithm to remove the unaligned sequences [42]. The Kimura 2-parameter (K2P) model was selected for constructing the *16S rRNA* and *gyrB* gene phylogenetic trees, respectively, using the Neighbor-Joining (NJ) and Maximum Likelihood (ML) methods (Bootstrap replication set to 1000) [43].

## Antibiotic selection and drug susceptibility test

A total of 13 antibiotics from 10 classes were selected for Kirby-Bauer disk diffusion susceptibility test: Penicillin G: amoxicillin (AMX, 30 ug); Chloramphenicol: florfenicol (FLR, 30 ug); Cephalosporin: cefradine (CED, 30 ug), cefuroxime (CXM, 30 ug), ceftazidime (CAZ, 30 ug),

cefepime (FEP, 30 ug); Monocyclic *β*-lactams: aztreonam (ATM, 30 ug); Carbapenem: mero-penem (MEM, 10 ug); Quinolone: ciprofloxacin (CIP, 5 ug); Sulfonamide: co-trimoxazole (SXT, 1.25/23.75 ug (Sulfamethoxazole (SMZ, 1.25 ug) / Trimethoprim (TMP, 23.75 ug)); Aminoglycoside: amikacin (AMK, 30 ug); Polypeptide: polymyxin B (POL, 300units); Tetracy-cline: tetracycline (TCY, 30 ug) [44, 45]. The above reagents are purchased from Hangzhou Microbial Regent Co., Ltd. (Hangwei, Hangzhou, China).

Quality control strains were selected from *Escherichia coli* ATCC 25922 and *Pseudomonas aeruginosa* ATCC 27853. The 102 *Aeromonas* isolates were inoculated on BHI (TOPBIO, Yan-tai, China) agar overnight and single-colony was picked and resuspended in 0.9% saline. The concentration of *Aeromonas* was adjusted to 1.5 x $10^8$ CFU/mL according to the McFarland scale, evenly coated on Muller Hinton Agar (Hopebio, Qingdao, China) and incubated at 28˚C for 18–24 hours in an inverted position. The diameter of the inhibition zone was recorded using vernier calipers. Refer to CLSI M45-A guidelines to determine the drug-resistant pheno-type of *Aeromonas* isolates (S1 Table) [32].

The diameter of the inhibition zone of different antibiotics was input into the resistance monitoring software WHONET 2020 to obtain the resistance phenotype of the isolates, the resistance rate of different antibiotics, the resistance spectrum (the kind of antibiotic the strain was simultaneously tolerant to), and the type of resistance spectrum (the different drug resis-tance profiles were named in a certain order) [46]. The diameter of inhibition zone was imported into BioNumerics software for cluster analysis, a cluster tree was drawn, and the spe-cies, host, sampling time, place, drug resistance spectrum, drug resistance profile type were added to this cluster tree.

## Drug-resistance genotype and phenotype analysis

Genomic DNA was extracted from 102 *Aeromonas* isolates using EasyPure® Genomic DNA Kit (TransGen Biotech, Beijing, China) as template and stored in -20˚C refrigerator. A variety of resistance gene primers were synthesized by reference: tetracycline resistance genes *tetA*, *tetB*, *tetE*; *β*-lactam resistance gene *TEM*; sulfonamide resistance genes *sul3*, *sul2*; chloram-phenicol resistance genes *catA7*, *floR*; aminoglycoside resistance genes *Ant(3")-I*; multi-drug transfer gene *qacEΔ1*, and quinolone resistance gene *oqxB* [47–51]. The PCR method was used to detect the carriage of various types of resistance genes under this resistance phenotype. Primer information is shown in S2 Table. The detection rate of drug resistant genes was counted, and the resistance phenotype and genotype match rate were calculated to speculate the reasons for the drug resistance of *Aeromonas* to different kinds of antibiotics.

## Multilocus sequence typing

Multilocus sequence typing (MLST) was conducted on 47 strains, comprising 33 multidrug-resistant *Aeromonas* obtained from prior experiments and 14 reference/type *Aeromonas* retrieved from the PubMLST database. Genomic DNA from the 33 multidrug-resistant *Aero-monas* was extracted using the EasyPure® Genomic DNA Kit (TransGen Biotech, Beijing, China), stored at -20˚C, and used as templates. The primers for the six housekeeping genes (*gyrB*, *groL*, *gltA*, *metG*, *ppsA*, and *recA*) employed in this study were sourced from a previous literature reference and synthesized by Limibio Biotechnology Co., Ltd. (Limibio, Hefei, China) [52]. Primer details are provided in S2 Table.

Amplification of target genes using polymerase chain reaction (PCR) was perforemd, using a 30 uL reaction mixture: 1.5 uL of genomic DNA template, 15 uL of MonAmp™ 2X Taq Mix (Monad, Wuhan, China), 1.5 uL each of forward and reverse primers, and 10.5 uL of ddH2O, resulting in a total volume of 30 uL. Amplification conditions: Initial denaturation at 94˚C for

5 min, followed by 30 cycles of 94˚C for 30 sec, 55–60˚C (Depending on the melting temperature of different primers) for 30 sec and 72˚C for 40 sec, and a final elongation at 72˚C for 5 min. Following amplification, 5 uL of the PCR product from each sample was subjected to 1% agarose gel electrophoresis at 120V for 30 min. Samples exhibiting specific bands were documented using a gel imaging system (Bio-Rad, CA, USA) and then sent to Limibio Biotechnology Co., Ltd. (Limibio, Hefei, China) for sequencing.

The allele sequences were submitted to the PubMLST website, and the allele numbers were assigned based on the comparison results. The allelic profiles of the six housekeeping genes for each strain were used to generate a unique allelic profile, which was assigned a sequence type (ST) number and concatenated according to the order of the genes on the chromosome, *gyrB* (477 bp), *groL* (510 bp), *gltA* (495 bp), *metG* (504 bp), *ppsA* (537 bp), and *recA* (561 bp), respectively. The concatenated sequences of these six genes, with a total length of 3084 bp, were analyzed in this study.

The genetic relatedness among the subject strains was investigated using the eBURST online tool (http://eburst.mlst.net/default.asp) to determine the presence of clonal complexes [53]. The eBURST analysis was conducted using various parameters, such as matching of 3–5 allele numbers in the *Aeromonas* MLST locus, assignment of STs into the same group, and clustering of strains with single (or double) locus differences into clonal complexes (CCs). The genomic relationships were further visualized using GrapeTree (https://pubmlst.org/GrapeTree), and the MLST data of the 33 multidrug-resistant *Aeromonas* were imported into the software to construct a sequence-based minimal spanning tree [54]. In addition, the data uploaded from China and all data available on the PubMLST website were analyzed separately to investigate the distribution of prevalent *Aeromonas* clones in China and worldwide.

PhyML 3.0 was used for phylogenetic analysis based on the maximum likelihood method [55]. Sequence comparison was performed using the Clustal Omega online tool (https://www.ebi.ac.uk/Tools/msa/clustalo/) with output format in phylip, and the file was imported into the PhyML online website (http://www.atgc-montpellier.fr/phyml/) for phylogenetic tree construction. The optimal replacement model was determined using the SMS tool (Smart Model Selection) based on the Bayesian Information Criterion (BIC), with the GTR model selected. The standard Bootstrap validation method was chosen, with the value set to 1000. The resulting phylogenetic tree was visualized using iTOL v6 [56].

Phylogenetic analysis was conducted using MEGA11 software. The MUSCLE algorithm was selected for sequence alignment, removal of unaligned sequences, trimming of redundant bases and gaps, and construction of a phylogenetic tree of the six housekeeping genes and concatenated genes using the Neighbor-Joining (NJ) method. The Kimura 2-parameter (K2P) model was selected with a bootstrap value of 2000. Additionally, a separate phylogenetic tree was constructed for each housekeeping gene and compared with the concatenated gene phylogenetic tree. The sequences of the six housekeeping genes were compared using BLAST on NCBI (https://blast.ncbi.nlm.nih.gov/Blast.cgi), and all the results were included in the phylogenetic tree as reference data. All 3084 bp concatenated sequences were translated into amino acid sequences using MEGA11 software, and the same method and parameters were used to construct the phylogenetic tree to compare the similarities and differences between the two.

## Recombination analysis

The concatenated nucleotide sequences of all 47 strains in this experiment were analyzed for the presence of recombination events using seven algorithms (RDP, GENCONV, BootScan, MaxChi, Chimaera, SiScan, and 3Seq) in the RDP v5.5 software [57]. A recombination event was considered to have occurred if it was supported by more than four algorithms. The clonal

relationships between isolates and possible recombination events were inferred using the maximum likelihood method with ClonalFrameML software, which detected the positions of recombination regions on each branch and constructed clonal relationship trees among the 47 strains in this study [58]. Use the R script cfml_results. R provided in ClonalFrameML (https://github.com/xavierdidelot/ClonalFrameML) to generate the graphical output of the analysis.

## Statistical analyses

The experimental data was analyzed using SPSS 20 software. Statistical analysis was performed using the Pearson correlation method to test the correlation coefficient between drug-resistance genotype and phenotype. The statistical significance level is set at a two-tailed α-value of 0.05, therefore, p-value less than 0.05 is considered statistically significant Multidrug-resistant strains are resistant to three or more antibiotics [45, 59]. The calculation formula of Multi antibiotic resistant MAR index was calculated by using the follow formula: MAR = Number of antibiotics to which an isolate showed resistance / Total number of antibiotics used in the experiment [60].

# Results

## Identification of *Aeromonas* species using *16S rRNA* and *gyrB* gene

In this study, we isolated total of 130 colonies from freshwater fish. Of the 130 samples, we sequenced 104 (80.00%) of the *16S rRNA* genes and 76 (58.46%) of the *gyrB* genes. The sequences were compared by BLAST, and were considered reliable if the similarity of sequencing results for the *16S rRNA* and *gyrB* gene ≥ 98% and ≥ 97%, respectively. As a result of confirming the highest score as the final result, the similarity interval after BLAST comparison was 98.95%-100% for the *16S rRNA* group and 97.54%-99.96% for the *gyrB* group. The identification results are as follows, for the *16S rRNA* group 76/104 *A. veronii* (73.08%), 12/104 *A. hydrophila* (11.54%), 9/104 *A. salmonicida* (8.65%), 5/104 *A. media* (4.81%), 1/104 *A. caviae* (0.96%), and 1/104 *A. rivipollensis* (0.96%); for the *gyrB* group 66/76 *A. veronii* (86.84%), 4/76 *A. hydrophila* (5.26%), 3/76 *A. salmonicida* (3.95%), 2/76 *A. media* (2.63%), and 1/76 *A. sobria* (1.32%) (Table 1). The identification results of the two groups showed that *A. veronii* was more predominant and might be the dominant *Aeromonas* species in freshwater fish farms in Hebei. In the *16S rRNA* group, *A. veronii* was detected in a lower proportion than in the *gyrB* group, and others were detected in a higher proportion and of more types in the *16S rRNA* group. This is because many alignments in the *16S rRNA* group showed 100% similarity and the same sequence score, *16S rRNA* genes may be more conserved.

**Table 1. Identification of the *16S rRNA* and *gyrB* gene.**

| Organism | Identification rate | |
|---|---|---|
| | *16S rRNA* gene group | *gyrB* gene group |
| *A. veronii* | 76/104 (73.08%) | 66/76 (86.84%) |
| *A. hydrophila* | 12/104 (11.54%) | 4/76 (5.26%) |
| *A. salmonicida* | 9/104 (8.65%) | 3/76 (3.95%) |
| *A. media* | 5/104 (4.81%) | 2/76 (2.63%) |
| *A. caviae* | 1/104 (0.96%) | - |
| *A. rivipollensis* | 1/104 (0.96%) | - |
| *A. sobria* | - | 1/76 (1.32%) |
| **Total** | 104/104 (100.00%) | 76/76 (100.00%) |

## Sequence analysis and phylogenetic analysis based on *16S rRNA* and *gyrB* gene

The matched sequences were imported into MEGA11 and DnaSP v6.12.03 to calculate a variety of sequence feature parameters to obtain sequence nucleotide diversity (Table 2). As shown in the Table 2, the number of the single polymorphic sites was significantly higher in *gyrB* group than the *16S rRNA* group. The segregating sites in the *gyrB* group accounted for 27.29%, much higher than 2.10% in the *16S rRNA* group, and the number of parsimony informative sites in the *gyrB* group was 12.61%, higher than 2.10% in the *16S rRNA* group. The average G + C content of the *gyrB* group was 59.2%, which was clearly distinguishable from the *16S rRNA* group of 55.8%.

Nucleotide diversity per site ($\pi$) and the average number of nucleotide differences per site ($\theta$) were significantly higher in the *gyrB* group compared to the *16S rRNA* group (Table 2). Where D value = 0, the actual variant type is similar to the predicted variant type; D value > 0, no or less rare alleles; D value < 0, more rare alleles. The *16S rRNA* group with a D value of 1.31 > 0 undergoes equilibrium selection or population contraction and population variation decreases, while the *gyrB* group with a D value of -1.55 < 0 undergoes population expansion and population variation increases. The ratio of the non-synonymous substitution rate to the synonymous substitution rate ($K_a/K_s$) is used to determine whether the gene is under selection pressure.

In this experiment, $K_a/K_s$ = 0.88 in the *16S rRNA* group tends to be close to 1, and is under natural selection pressure, indicating that the *16S rRNA* genes are quite stable in the evolutionary process. In contrast, $K_a/K_s$ = 0.036 in the *gyrB* group was under purifying selection, indicating that the *gyrB* gene tended to specialize during evolution. $K_a/K_s$ was consistent with the results of Tajima's D test and the results of polymorphic loci and parsimony informative loci, all of which proved that the *gyrB* gene had higher diversity, and therefore the *gyrB* gene was more suitable for interspecific identification of *Aeromonas* spp.

We reconstructed phylogenetic trees using Neighbor-Joining (NJ) and Maximum Likelihood (ML) methods. As shown in Fig 1A and 1B, four *A. veronii* strains (21619, 21618–2, 2021wx5, and 21616) clustered with *A. salmonicida* reference strain in the *16S rRNA* phylogenetic tree (Fig 1A and 1B). *A. salmonicida* strain 20813 clustered with *A. veronii* as a single clade. *A. media*

**Table 2. Nucleotide diversity of *Aeromonas* isolates in this study.**

| Parameters | *16S rRNA* Group | *gyrB* Group |
|---|---|---|
| Number of the sites | 1334 | 634 |
| Single polymorphic site | 3 | 93 |
| Missing/gap | 2 | 2 |
| Segregating sites (%) | 28 (2.10) | 173 (27.29) |
| Parsimony informative sites (%) | 28 (2.10) | 80 (12.61) |
| G+C content | 0.5580 | 0.5920 |
| $\pi$ | 0.0066 | 0.0382 |
| $\theta$ | 0.0047 | 0.0654 |
| Tajima's D | 1.3162 | -1.5518 |
| $K_a$ | 0.0060 | 0.0049 |
| $K_s$ | 0.0068 | 0.1372 |
| $K_a/K_s$ | 0.8770 | 0.0359 |

Note: $\pi$, nucleotide diversity per site; $\theta$, average number of nucleotide differences per site; $K_a$, number of nonsynonymous changes per nonsynonymous site; $K_s$, number of synonymous changes per synonymous site.

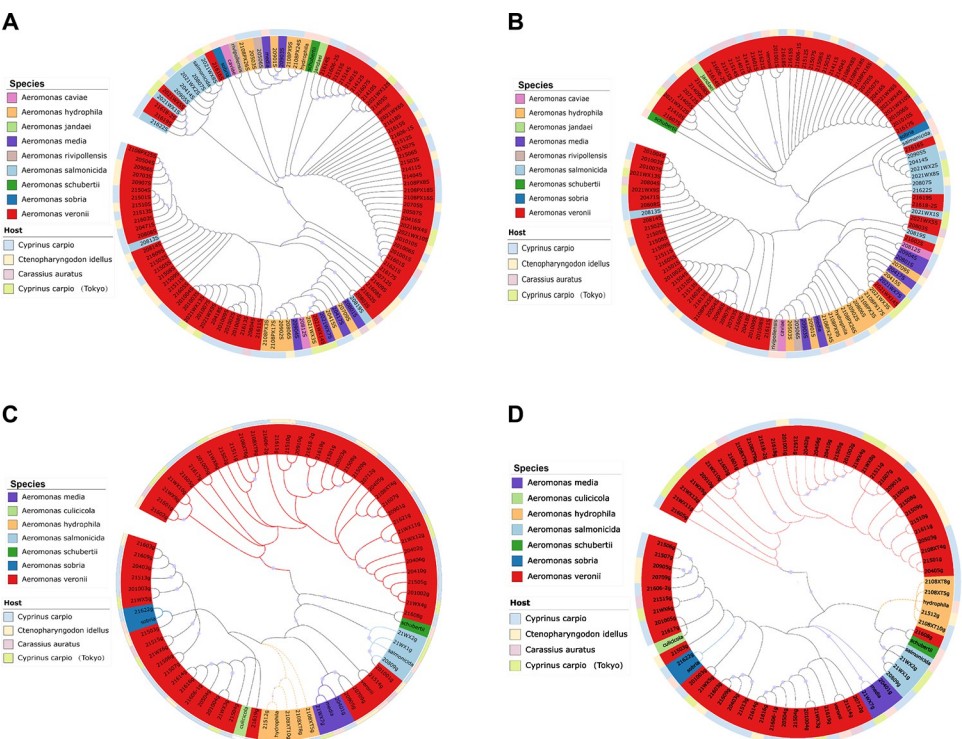

**Fig 1. Circular phylogenetic tree of *16S rRNA* and *gyrB* gene.** The phylogenetic tree generated by the MEGA11 software was modified using iTOL v6 software to add the corresponding sampling time and host information for each sequence, and to facilitate the distinction between the two genes, *gyrB* group strain name followed by the letter g and *16S rRNA* group added the letter S. The lavender dots represent the bootstrap value size. **(A)** Neighbor-joining phylogenetic tree of *16S rRNA* gene. **(B)** Maximum Likelihood phylogenetic tree of *16S rRNA* gene. **(C)** Neighbor-joining phylogenetic tree of *gyrB* gene. **(D)** Maximum Likelihood phylogenetic tree of *gyrB* gene.

and *A. rivipollensis* reference strains clustered together with the *Aeromonas hydrophila* branch. No significant associations were found between strains and hosts in either phylogeny, suggesting that *Aeromonas* can be commonly found in various freshwater fish species.

In the *gyrB* phylogeny, each strain was clustered into one group with its corresponding reference strain. Strain 21622 was identified and clustered with *A. sobria*, but was clustered with *A. salmonicida* in the *16S rRNA* phylogenetic trees (Fig 1). Strains 2021WX8 and 2021WX3 were classified as *A. salmonicida* and *A. hydrophila*, respectively, in the *16S rRNA* phylogenetic trees, but clustered with *A. veronii* in the *gyrB* phylogenetic trees. Strain 20901, 20709, and 20503 were classified as *A. hydrophila* in the *16S rRNA* phylogenetic trees, but clustered with *A. veronii* in the *gyrB* phylogenetic trees. Interestingly, *A. schubertii* clustered with *A. veronii* in two different sets of trees. The reference strain of *A. culicicola* clusters with *A. veronii*, supporting two reference strains being considered synonymous.

## Antibiotic susceptibility test and drug resistance profile

For phenotypic analysis of antibiotic susceptibility and drug resistance, we measured the zone inhibition using 13 common antibiotics on 102 fully cultured *Aeromonas* isolates according to the CLSI-2020 guidelines. The rate of drug resistance to amoxicillin was the highest (100%), followed by tetracycline (18.6%) and cefradine (13.7%), The proportion of moderate resistance to co-trimoxazole, meropenem and florfenicol was 4.9%, 8.8% and 9.8%, respectively. It is sensitive to cefuroxime, ceftazidime, cefepime, aztreonam, and polymyxin B (Table 3).

**Table 3. Antibiotic and drug resistance profile.**

| Antimicrobial category | Antimicrobial agent | Results of antimicrobial susceptibility test | | |
|---|---|---|---|---|
| | | R (%) | I (%) | S (%) |
| Penicillins | Amoxicillin (AMX) | 100 | 0.0 | 0.0 |
| Chloramphenicols | Florfenicol (FLR) | 9.8 | 1.0 | 89.2 |
| Cephalosporins | Cefradine (CED) | 13.7 | 58.8 | 27.5 |
| | Cefuroxime (CXM) | 0.0 | 1.0 | 99.0 |
| | Ceftazidime (CAZ) | 0.0 | 0.0 | 100.0 |
| | Cefepime (FEP) | 0.0 | 0.0 | 100.0 |
| Monocyclic β-lactams | Aztreonam (ATM) | 2.0 | 1.0 | 97.0 |
| Carbapenems | Meropenem (MEM) | 8.8 | 19.6 | 71.6 |
| Quinolones | Ciprofloxacin (CIP) | 0.0 | 2.9 | 97.1 |
| Sulfonamides | Co-trimoxazole (SXT) | 4.9 | 12.7 | 82.4 |
| Aminoglycosides | Amikacin (AMK) | 2.9 | 1.0 | 96.1 |
| Polypeptides | Polymyxin B (POL) | 1.0 | 2.0 | 97.0 |
| Tetracyclines | Tetracycline (TCY) | 18.6 | 1.0 | 80.4 |

The analysis of the antimicrobial resistance spectrum of *Aeromonas* showed resistance to 11 of the 13 tested antibiotics, with ceftazidime and cefepime showing no resistance or moderate susceptibility. The resistance spectrum analysis of 102 *Aeromonas* isolates showed that they are resistant to at least one and up to seven antibiotics (Table 4). The range of multi antibiotic resistant (MAR) index was from 0.08 to 0.54. The two isolates with the highest MAR index are strain 201006 and 21502 which exhibited resistant to seven antibiotics. The MAR index of 18 strains was 0.08, indicating these strains were resistant to at least one type of antibiotic. 44 out of 102 *Aeromonas* isolates were resistant to at least three antibiotics, and the MAR index was higher than 0.23.

A total of 26 resistance spectrum (designated A-Z) were observed in the 102 *Aeromonas* isolates. The most prevalent resistance spectrum was Type B (CED/AMX) which accounted for 32.35% of the isolates, followed by type A (AMX), which accounted for 17.6%. All 102 *Aeromonas* isolates were resistant to amoxicillin, and 40/102 isolates (39.21%) were resistant to two antibiotics. Of 102 isolates, 44 isolates (43.14%) were phenotypically multidrug-resistant bacteria with resistant to three or more antibiotics: 24/102 (23.53%) for three antibiotics, 10/102 (9.80%) for four antibiotics, 5/102 (4.90%) for five antibiotics, 3/102 (2.94%) for six antibiotics, and 2/102 (1.96%) for seven antibiotics (Table 4). Eleven isolates (10.78%) were phenotypically resistant to the most common multidrug-resistant spectrums, type K (MEM/CED/AMX) spectrum, followed by five isolates (4.9%) each in the type G (TCY/CED/AMX) and M (MEM/TCY/CED/AMX) spectrums (Table 4).

Clustering tree analysis was performed on the antimicrobial resistance spectrum of each strain by integrating information on strain origin, bacterial species, antimicrobial resistance spectrum, and type of resistance spectrum. The clustering tree analysis divided into two main branches: the first branch, consisted of 23 resistance spectrums, and the second branch, consisted of 3 resistance spectrums (Fig 2). Branch I contained 82 isolates, which were subdivided into four sub-branches (A, B, C, and D). Ten of the 82 isolates belong to sub-branch A, with nine types of resistance spectrums: type V (FLR/ATM/TCY/CED/AMX), Z (FLR/MEM/CIP/SXT/TCY/CED/AMX), X (FLR/MEM/SXT/TCY/CED/AMX), P (FLR/TCY/CED/AMX), Q (FLR/SXT/CED/AMX), U (FLR/MEM/SXT/CED/AMX), R (FLR/MEM/CED/AMX), L (FLR/CED/AMX), and W (FLR/ATM/MEM/CED/AMX). The diversity of the resistance spectrum in this sub-branch A is relatively affluent, with type Q and type U clustering together and Type

**Table 4. Drug resistance spectrum of 102 *Aeromonas* isolates.**

| Strains | Resistance Spectrum | Type | MDR | MAR | Amount | Percentage (%) |
|---|---|---|---|---|---|---|
| WX8, WX13, WX10, 216182, 21613, 21609, 21603, 21602, 21514, 21513, 21512, 21411, 21404, 21403, 20812, 20808, 20418, 20411 | AMX | A | 0 | 0.08 | 18/102 | 17.65 |
| WX7, WX5, WX3, WX1, 216062, 216061, 201008, 21618, 21615, 21611, 21608, 21605, 21508, 21505, 21504, 21503, 21407, 21405, 21401, 20904, 20903, 20902, 20815, 20814, 20813, 20803, 20709, 20616, 20611, 20506, 20505, 20414, 201009, | CED/AMX | B | 0 | 0.15 | 33/102 | 32.35 |
| 20710 | TCY/AMX | C | 0 | 0.15 | 1/102 | 0.98 |
| 21501, 20417 | SXT/AMX | D | 0 | 0.15 | 2/102 | 1.96 |
| 20507 | CIP/AMX | E | 0 | 0.15 | 1/102 | 0.98 |
| 21506, 20907, 20906 | MEM/AMX | F | 0 | 0.15 | 3/102 | 2.94 |
| 21601, 21511, 20905, 20804, 20712 | TCY/CED/AMX | G | 5 | 0.23 | 5/102 | 4.9 |
| WX14, 20416 | POL/CED/AMX | H | 2 | 0.23 | 2/102 | 1.96 |
| 21616, 21515, 20809, 20415 | SXT/CED/AMX | I | 4 | 0.23 | 4/102 | 3.92 |
| 201001 | SXT/TCY/AMX | J | 1 | 0.23 | 1/102 | 0.98 |
| 201010, 21621, 21619, 21510, 21509, 21507, 21409, 20910, 20801, 20705, 20617 | MEM/CED/AMX | K | 11 | 0.23 | 11/102 | 10.78 |
| 20703 | FLR/CED/AMX | L | 1 | 0.23 | 1/102 | 0.98 |
| WX6, 201007, 201004, 20901, 20503 | MEM/TCY/CED/AMX | M | 5 | 0.31 | 5/102 | 4.9 |
| 201003 | MEM/SXT/TCY/AMX | N | 1 | 0.31 | 1/102 | 0.98 |
| 21607 | MEM/SXT/AMK/AMX | O | 1 | 0.31 | 1/102 | 0.98 |
| 201002 | FLR/TCY/CED/AMX | P | 1 | 0.31 | 1/102 | 0.98 |
| 21410 | FLR/SXT/CED/AMX | Q | 1 | 0.31 | 1/102 | 0.98 |
| 21408 | FLR/MEM/CED/AMX | R | 1 | 0.31 | 1/102 | 0.98 |
| 21614 | SXT/AMK/CXM/CED/AMX | S | 1 | 0.39 | 1/102 | 0.98 |
| 21412 | MEM/SXT/POL/CED/AMX | T | 1 | 0.39 | 1/102 | 0.98 |
| 20471 | FLR/MEM/SXT/CED/AMX | U | 1 | 0.39 | 1/102 | 0.98 |
| 21402 | FLR/ATM/TCY/CED/AMX | V | 1 | 0.39 | 1/102 | 0.98 |
| 21617 | FLR/ATM/MEM/CED/AMX | W | 1 | 0.39 | 1/102 | 0.98 |
| 21622, 21406, 20504 | FLR/MEM/SXT/TCY/CED/AMX | X | 3 | 0.46 | 3/102 | 2.94 |
| 201006 | MEM/CIP/SXT/AMK/TCY/CED/AMX | Y | 1 | 0.54 | 1/102 | 0.98 |
| 21502 | FLR/MEM/CIP/SXT/TCY/CED/AMX | Z | 1 | 0.54 | 1/102 | 0.98 |

R and type L forming another cluster. Notably, Strain 21502 (Z) and Strain 21402 (V) cluster together due to their resistance spectrum containing four antibiotics (FLR/TCY/CED/AMX). Strain 21617 (W) is remarkable as it is the only isolate in sub-branch A simultaneously resistant to ampicillin and meropenem.

Ten isolates belong to sub-branch B, which has four types of resistance spectrums, including CED/AMX antibiotics: type B (CED/AMX), H (POL/CED/AMX), K (MEM/CED/AMX), and T (MEM/SXT/POL/CED/AMX). Interestingly, Strain 20910 is phenotypically resistant to Type K spectrum, but differs from other type K strains belonging to sub-branch C in that due to moderately sensitive to meropenem and resistant to cefradine and amoxicillin.

In sub-branch C, 60 isolates are contained with 12 resistance spectrums: type M (MEM/TCY/CED/AMX), G (TCY/CED/AMX), F (MEM/AMX), K (MEM/CED/AMX), B (CED/

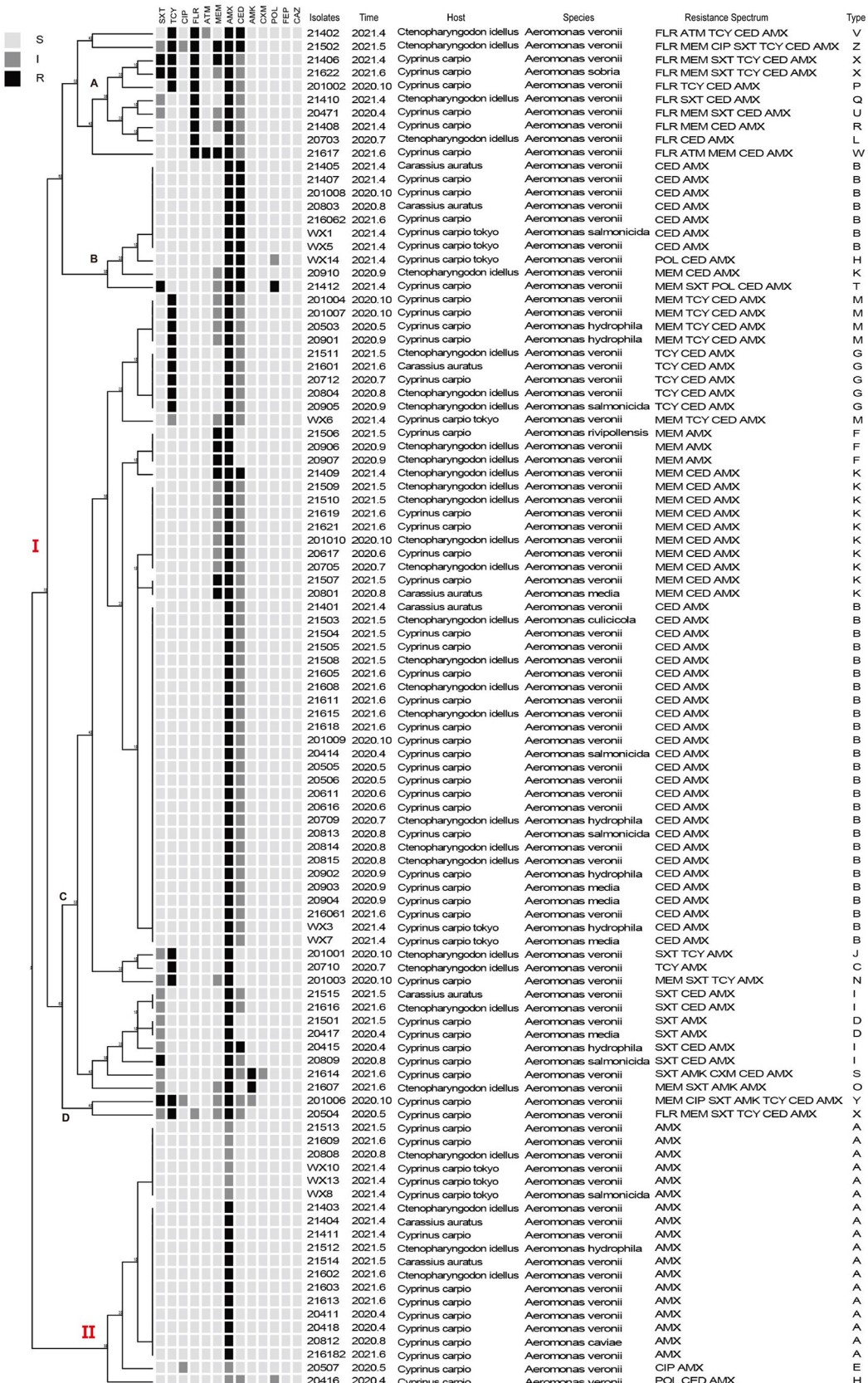

**Fig 2. Cluster tree of 102 drug-resistance *Aeromonas* isolates.** The cluster tree was generated by the BioNumerics software. Clustering tree analysis was performed on the antimicrobial resistance spectrums of each bacterial strain, and information on strain origin, bacterial species, antimicrobial resistance spectrums, and resistance spectrums (Type) were incorporated into the clustering tree.

AMX), J (SXT/TCY/AMX), C (TCY/AMX), N (MEM/SXT/TCY/AMX), I (SXT/CED/AMX), D (SXT/AMX), S (SXT/AMK/CXM/CED/AMX), and O (MEM/SXT/AMK/AMX). Sub-branch C is the most diverse regarding the presence of resistant spectrums compared to the other sub-branches. Strains with Type B, D, F, and G cluster together due to their identical resistance phenotypes, while strains with Type S, O, and N cluster separately because of their distinct resistance spectrums. Four strains share Type I but are separated into three clusters based on different resistance levels and sensitivity to the antibiotics in the resistance spectrum. Strain WX6 is distinguished from other Type M strains as it exhibits moderate sensitivity to tetracycline, whereas all other Type M strains are resistant. Strain 20801 and 21507 cluster together in Type K because they exhibit resistance to meropenem, while all other Type K strains show only moderate sensitivity.

Sub-branch D contained two isolates with two types of resistance spectrums: type X (FLR/MEM/SXT/TCY/CED/AMX) and Y (MEM/CIP/SXT/AMK/TCY/CED/AMX). The strain 20504 is resistant only to tetracycline and amoxicillin and moderately sensitive to other antibiotics, clearly distinguishing it from type K strains belonging to sub-branch A.

Branch II contained 20 isolates with three resistance spectrums: type A (AMX), type E (CIP/AMX), and H (POL/CED/AMX). Type A was clustered into for amoxicillin moderately sensitive strains and amoxicillin-resistant, respectively. Type E (CIP/AMX) and type H (POL/CED/AMX) contains two rare antibiotics, ciprofloxacin and polymyxin B, respectively.

## Drug-resistant phenotypes of the recovered isolates

The relationship between drug resistance and species of isolates was analyzed by using the WHONET software. The drug-resistant phenotype of *A. sobria* was type X according to the spectrum; The drug-resistant phenotype of *A. caviae* was type A; *A. salmonicida* contains four drug-resistant phenotypes type A, B, G, and I; *A. hydrophila* contained four phenotypes, type A, B, I and M; *A. veronii* contains all spectrum type A-Z. It can be seen that there is a large difference in the number of samples. Except for type A and B, the frequency of other drug-resistant phenotypes has no obvious correlation with strains.

## Drug-resistance genotype and phenotype analysis

We used 11 drug resistance genes in 7 classes to analyze the relationship between genotypic and phenotypic drug resistance profiles in 102 isolates. Out of 11, eight genes were successfully detected but three (*sul3*, *catA7*, and *oqxB*) were not detected. The detection rates of different drug resistance genes were different: *TEM* (100%), *tetA* (75%), *floR* (72.7%), *sul2* (50%), *tetE* (40%), *tetB* (30%), *qacE△1* (27.2%), and *Ant(3")*-I (22.7%) (Table 5).

*TEM* gene was detected in all 102 isolates, consistent with the previously speculated natural resistance of *Aeromonas* to amoxicillin. Of the 18 isolates showing a resistance spectrum of resistance to CO trimoxazole (SXT), the *sul2* gene was detected in nine (201001, 20471, 21501, 21607, 201006, 20809, 21406, 21412, 21622) isolates, demonstrating a 50% concordance between drug-resistance phenotype and genotype. *floR* gene was amplified in eight of 11 isolates, representing a spectrum of resistance to florfenicol (FLR), resulting in 72.7% concordance between drug-resistance phenotype and genotype. In addition, among the 20 isolates with a tetracycline resistance spectrum, 75% (15/20), 30% (6/20), and 40% (8/20) of the *tetA*,

**Table 5. Detection rate of drug-resistance gene.**

| Resistance | Gene | Detections/Total | Detection rate |
|---|---|---|---|
| Tetracycline | *tetA* | 15/20 | 75.0% |
| | *tetB* | 6/20 | 30.0% |
| | *tetE* | 8/20 | 40.0% |
| *β*-lactam | TEM | 102/102 | 100.0% |
| Sulfonamide | *sul3* | 0/18 | 0.0% |
| | *sul2* | 9/18 | 50.0% |
| Chloramphenicol | *catA7* | 0/11 | 0.0% |
| | *floR* | 8/11 | 72.7% |
| Aminoglycoside | *Ant(3″)-I* | 10/44 | 22.7% |
| Multi-drug transfer | *qacE△1* | 12/44 | 27.2% |
| Quinolone | *oqxB* | 0/2 | 0.0% |

*tetB*, and *tetE* genes were consistent between phenotypes and genotypes, respectively. Of the 20 isolates, only the WX6 strain was undetectable for any of the tetracycline resistance genes (S3 Table).

Ten of 44 multidrug-resistant strains were detected in *Ant(3″)-I* (20471, 20503, 201002, 201006, 21406, 21410, 21502, 21607, 21614 and WX6). Among them, three isolates (201006, 21607 and 21614) were resistant to amikacin (AMK) in the drug sensitivity test, and the *Ant (3″)-I* gene was detected in these isolates, demonstrating a 100% concordance between drug-resistance phenotype and genotype. The drug resistance phenotype and genotype were not completely consistent. The cluster analysis of the drug resistance profile showed that the phenotype of the isolates was not significantly correlated with factors such as the sampling year, location, host, species and other factors. This suggests that the environmental selection pressure may cause bacterial drug resistance to change over time.

## Multilocus sequence typing of multidrug-resistant *Aeromonas*

To investigate the genetic correlations, six housekeeping genes (*gyrB*, *groL*, *gltA*, *metG*, *ppsA*, *recA*) were analyzed in a total 47 strains, containing 33 multidrug-resistant *Aeromonas* isolates and 14 reference strains. Among the 33 isolates, the six housekeeping genes and the concatenated genes showed a remarkable abundance of polymorphic and parsimony informative sites (Table 6). Polymorphic sites ranged from a minimum of 26.83% for the *gyrB* gene to

**Table 6. Nucleotide diversity of all isolates included in this study.**

| Gene | L | S | P | G+C | $\pi$ | $\theta$ | Tajima's D | $K_a/K_s$ | Syn change | Nonsyn change |
|---|---|---|---|---|---|---|---|---|---|---|
| *gyrB* | 477 | 128 | 79 | 0.596 | 0.0608 | 0.0440 | -0.9940 | 0.0202 | 148 | 10 |
| *groL* | 510 | 170 | 130 | 0.586 | 0.0755 | 0.0618 | -0.6569 | 0.0405 | 158 | 8 |
| *gltA* | 495 | 139 | 99 | 0.603 | 0.0636 | 0.0523 | -0.6382 | 0.0634 | 139 | 10 |
| *metG* | 504 | 154 | 117 | 0.577 | 0.0692 | 0.0546 | -0.7621 | 0.0704 | 132 | 10 |
| *ppsA* | 537 | 192 | 138 | 0.635 | 0.0810 | 0.0766 | -0.1958 | 0.0590 | 179 | 25 |
| *recA* | 561 | 163 | 102 | 0.597 | 0.0660 | 0.0501 | -0.8732 | 0.0532 | 140 | 35 |
| Concatenated gene | 3084 | 937 | 664 | 0.600 | 0.0688 | 0.05640 | -0.6620 | 0.0451 | 927 | 72 |

Note: L, length (bp) of the sequences; S, number of polymorphic sites; P, parsimony informative sites; $\pi$, nucleotide diversity per site; $\theta$, average number of nucleotide differences per site; $K_a$, number of nonsynonymous changes per nonsynonymous site; $K_s$, number of synonymous changes per synonymous site.

a maximum of 35.75% for the *ppsA* gene, while parsimony informative sites varied from a minimum of 16.56% for the *gyrB* gene to a maximum of 25.70% for the *ppsA* gene. The G+C content showed slight variations among the housekeeping genes, ranging from 57.7% (*metG*) to 63.5% (*ppsA*). Nucleotide diversity ($\pi$) values spanned from 0.0608 (*gyrB*) to 0.0810 (*ppsA*). Mean nucleotide differences ($\theta$) per site were distributed across a range of 0.0440 (*gyrB*) to 0.0766 (*ppsA*). The high values observed for both $\pi$ and $\theta$ indicate a substantial level of polymorphism in all gene sequences in this study. The genetic equilibrium of the alleles was analyzed using Tajima's D neutrality test, which showed that all D values were less than 0, indicating population expansion and increased population variability. The distribution of $K_a$/$K_s$ values ranged from 0.0202 (*gyrB*) to 0.0704 (*metG*), and all seven gene sequences had much less than 1, indicating purifying selection as the selective pressure and demonstrating that the concatenated genes are suitable for population studies.

The ratio of synonymous substitutions to nonsynonymous substitutions was much higher, with the *groL* gene having the highest ratio of 19.75-fold and the *recA* gene having the lowest ratio of 4-fold, suggesting that the six selected housekeeping genes are also suitable for population studies. The concatenated genes exhibited an average of 937 polymorphic sites, and other diversity parameters consistently displayed high values, consistent with the results of the six housekeeping genes.

To obtain MLST typing data, MLST method was used to by processing concatenated sequences of 33 multidrug-resistant *Aeromonas* and submitting them to the PubMLST website. A total 47 strains, including 14 reference strains were integrated and analyzed (Table 7). A total of 43 sequence types (ST) were identified in 47 strains, of which 30 new STs (ST1024-ST1053) for the *gyrB*, *gltA*, *groL*, *metG*, *ppsA*, and *recA* genes were found in 33 strains. The six strains were classified into three STs, two each. 20712 and 20705 were assigned to ST1042, 20503 and 20901 to ST1030, and 21409 and 20801 to ST1026 (Table 7).

Homologous recombination was a significant complicating factor when analyzing phylogenetic relationships among closely related genotypes, exerting a substantial influence on sequence analysis. The number of locus variants (nLV) at each genetic locus was determined utilizing the eBURST algorithm(http://eburst.mlst.net), facilitating the establishment of connections between sequence types within the constructed minimum spanning tree (Fig 3). Considering single locus variants (SLV), only two strains, ST1027 and ST1047, formed a clonal complex, when single locus variants (SLV) were considered (Fig 3A). Considering double-locus variants (DLVs) and triple-locus variants (TLVs), the number increased to four and five clonal complexes, respectively (Fig 3B and 3C). These observations suggest that 33 multidrug-resistant *Aeromonas* are likely to have distant genetic relationships with no obvious clonal connections.

To further investigate the genetic diversity, we integrated data from all isolates in China available into the PubMLST database. We analyzed 524 isolates and found that only 36 clonal complexes were present. Interestingly, only one central sequence type (ST94) was identified in this dataset. This expanded dataset included eight central sequence types (ST94, ST267, ST377, ST609, ST745, ST750, ST751, and ST755), and we Performed eBURST analysis on all STs uploaded to the database. Consistent with our analysis of the 33 multidrug-resistant *Aeromonas* in this experiment, we found that the clonal groups with highly prevalent did not clustered among the *Aeromonas* strains in the database.

## Gene phylogenetic analysis based on concatenated and housekeeping gene

To estimate the phylogenetic relationships among the isolates, the concatenated six housekeeping gene sequences derived form 47 strains was constructed phylogenetic trees using two

**Table 7. Origins and MLST data of *Aeromonas* strains analyzed in this study.**

| Strain | Nation | Source | Species | *gltA* | *groL* | *gyrB* | *metG* | *ppsA* | *recA* | ST |
|---|---|---|---|---|---|---|---|---|---|---|
| NCIMB 882 | Argentina | Carassius auratus | *A. caviae* | 4 | 3 | 4 | 3 | 3 | 3 | 3 |
| DSM 11577[T] | Spain | Anguilla anguilla | *A. encheleia* | 8 | 7 | 8 | 7 | 7 | 7 | 7 |
| NCIMB 1134 | UK | Rainbow trout | *A. hydrophila* | 5 | 4 | 5 | 4 | 4 | 4 | 4 |
| CECT 4228[T] | USA | Diarrhea | *A. jandaei* | 15 | 14 | 15 | 14 | 14 | 14 | 14 |
| DSM 4881[T] | UK | Fish farm effluent | *A. media* | 7 | 6 | 7 | 6 | 6 | 6 | 6 |
| DSM 19604T | Belgium | Other | *A. popoffii* | 11 | 10 | 11 | 10 | 10 | 10 | 10 |
| NCIMB 1102[T] | UK | Salmon | *A. salmonicida* | 2 | 2 | 2 | 2 | 2 | 2 | 2 |
| CECT 4240[T] | USA | Wounds | *A. schubertii* | 16 | 15 | 16 | 15 | 15 | 15 | 15 |
| CECT 4245[T] | France | Fish | *A. sobria* | 21 | 20 | 21 | 20 | 18 | 20 | 19 |
| CECT 4255[T] | India | Stools | *A. trota* | 18 | 17 | 18 | 17 | 16 | 17 | 16 |
| CECT 4257[T] | USA | Nasopharyngeal secretion | *A. veronii* | 19 | 18 | 19 | 18 | 16 | 18 | 17 |
| CECT 4199[T] | Spain | Anguilla anguilla | *A. allosaccharophila* | 14 | 13 | 14 | 13 | 13 | 13 | 13 |
| CECT 7083[T] | Spain | Other | *A. tecta* | 164 | 168 | 155 | 164 | 169 | 164 | 186 |
| CECT 5864[T] | Spain | Mollusc | *A. molluscorum* | 165 | 169 | 156 | 165 | 170 | 165 | 187 |
| WX6 | China | Cyprinus carpio | *A. veronii* | 110 | 694 | 733 | 307 | 140 | 361 | 1024 |
| 21402 | China | Ctenopharyngodon idellus | *A. veronii* | 765 | 273 | 128 | 659 | 511 | 775 | 1025 |
| 21409 | China | Ctenopharyngodon idellus | *A. veronii* | 736 | 239 | 733 | 48 | 344 | 653 | 1026 |
| 21410 | China | Ctenopharyngodon idellus | *A. veronii* | 558 | 295 | 87 | 222 | 232 | 595 | 1027 |
| 20471 | China | Cyprinus carpio | *A. veronii* | 558 | 300 | 764 | 222 | 232 | 458 | 1028 |
| 21502 | China | Cyprinus carpio | *A. veronii* | 350 | 342 | 348 | 531 | 797 | 371 | 1029 |
| 20503 | China | Cyprinus carpio | *A. veronii* | 80 | 215 | 356 | 79 | 236 | 82 | 1030 |
| 20504 | China | Cyprinus carpio | *A. veronii* | 775 | 490 | 479 | 735 | 718 | 53 | 1031 |
| 20507 | China | Ctenopharyngodon idellus | *A. veronii* | 646 | 460 | 733 | 112 | 304 | 532 | 1032 |
| 21509 | China | Ctenopharyngodon idellus | *A. veronii* | 70 | 215 | 235 | 79 | 236 | 82 | 1033 |
| 21510 | China | Ctenopharyngodon idellus | *A. veronii* | 701 | 122 | 633 | 780 | 810 | 801 | 1034 |
| 21511 | China | Ctenopharyngodon idellus | *A. veronii* | 578 | 5 | 635 | 735 | 691 | 773 | 1035 |
| 21515 | China | Carassius auratus | *A. veronii* | 705 | 634 | 242 | 737 | 66 | 784 | 1036 |
| 21601 | China | Carassius auratus | *A. veronii* | 726 | 24 | 87 | 685 | 797 | 476 | 1037 |
| 21607 | China | Ctenopharyngodon idellus | *A. veronii* | 726 | 551 | 87 | 531 | 642 | 326 | 1038 |
| 21614 | China | Ctenopharyngodon idellus | *A. veronii* | 726 | 24 | 87 | 175 | 642 | 476 | 1039 |
| 21617 | China | Cyprinus carpio | *A. veronii* | 134 | 140 | 128 | 134 | 718 | 134 | 1040 |
| 21622 | China | Cyprinus carpio | *A. veronii* | 300 | 158 | 326 | 574 | 642 | 182 | 1041 |
| 20705 | China | Ctenopharyngodon idellus | *A. veronii* | 56 | 117 | 115 | 117 | 114 | 112 | 1042 |
| 20712 | China | Ctenopharyngodon idellus | *A. veronii* | 56 | 117 | 115 | 117 | 114 | 112 | 1042 |
| 20801 | China | Carassius auratus | *A. veronii* | 736 | 239 | 733 | 48 | 344 | 653 | 1026 |
| 20804 | China | Ctenopharyngodon idellus | *A. veronii* | 120 | 117 | 115 | 117 | 124 | 86 | 1043 |
| 20901 | China | Cyprinus carpio | *A. veronii* | 80 | 215 | 356 | 79 | 236 | 82 | 1030 |
| 20905 | China | Ctenopharyngodon idellus | *A. veronii* | 129 | 289 | 111 | 141 | 810 | 801 | 1044 |
| 20910 | China | Ctenopharyngodon idellus | *A. veronii* | 140 | 694 | 633 | 531 | 691 | 365 | 1045 |
| 201001 | China | Ctenopharyngodon idellus | *A. veronii* | 152 | 694 | 145 | 145 | 156 | 151 | 1046 |
| 201002 | China | Cyprinus carpio | *A. veronii* | 558 | 639 | 87 | 222 | 232 | 595 | 1047 |
| 201003 | China | Cyprinus carpio | *A. veronii* | 358 | 221 | 356 | 780 | 810 | 383 | 1048 |
| 201004 | China | Cyprinus carpio | *A. veronii* | 120 | 117 | 22 | 780 | 124 | 112 | 1049 |
| 201006 | China | Cyprinus carpio | *A. veronii* | 467 | 700 | 40 | 141 | 511 | 784 | 1050 |
| 201007 | China | Cyprinus carpio | *A. veronii* | 761 | 182 | 722 | 685 | 691 | 183 | 1051 |
| 201009 | China | Cyprinus carpio | *A. veronii* | 740 | 289 | 111 | 780 | 797 | 801 | 1052 |
| 201010 | China | Ctenopharyngodon idellus | *A. veronii* | 124 | 117 | 733 | 117 | 114 | 112 | 1053 |

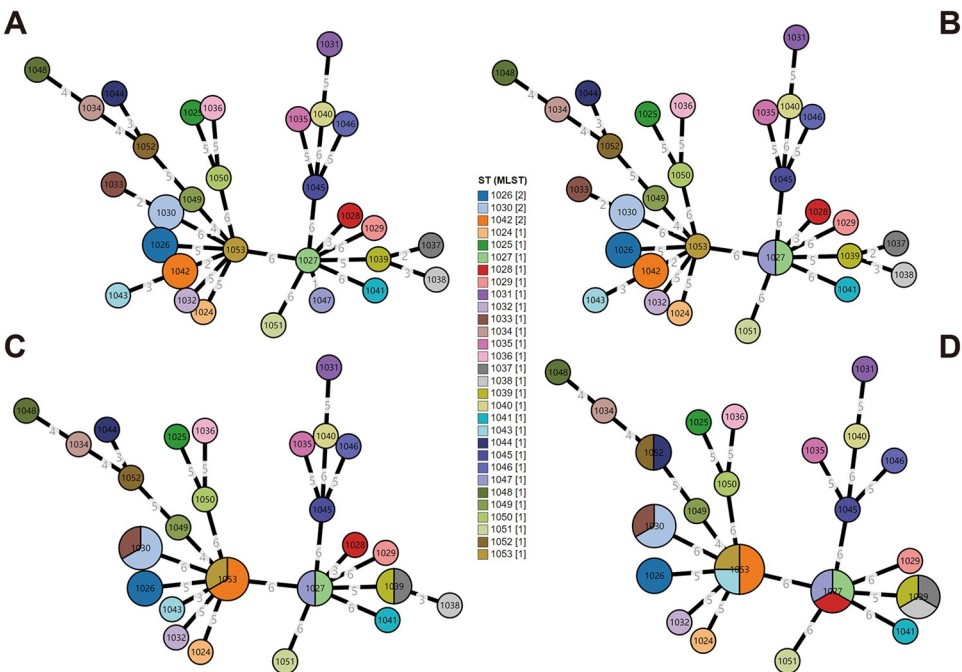

**Fig 3. GrapeTree of 33 multidrug-resistant *Aeromonas* isolates based on n locus variant.** GrapeTree of the minimum spanning tree was based on n = 0, 1, 2, 3. The numbers accompanying the sequence types in the legend denote their respective frequencies, while the sizes of the circles represent their kurtosis values. Furthermore, the numbers present on the connecting lines signify the locus differences. **(A)** This shows the original minimum spanning tree, where n = 0 and none of the 30 STs are clustered together, and the least different types are ST1027 and ST1047, which differ by only one locus. **(B)** This shows the SLV (single locus variant) minimum spanning tree, where ST1027 and ST1047 form a clonal complex. **(C)** This shows the DLV (double locus variant) minimum spanning tree, where ST1053 and ST1042, ST1030 and ST1033, ST1037 and ST1039 merge. **(D)** This shows the TLV (triple locus variant) minimum spanning tree, where ST1044 clusters with ST1052, ST1028 clusters into ST1027, ST1043 clusters into ST1053, and ST1038 clusters into ST1029.

methods (NJ and ML). A phylogenetic tree using PhyML with ML method comprises three primary branches (Branch I, II, and III). Two primary branches (Branch I and II) contain only DSM 4881 and NCIMB 882 strains, respectively. In comparison, Branch III clusters 33 multidrug-resistant *Aeromonas* with the reference strain CECT 4257[T] of *A. veronii*, strain CECT 4245[T] of *A. sobria*, and CECT 4199[T] of *A. allosaccharophila* (Fig 4).

A phylogenetic tree with NJ was constructed by MEGA11 using the concatenated nucleotide sequences (3084 bp) and amino acid sequences of the six housekeeping genes from 47 strains. Additionally, each nucleotide sequence of six housekeeping genes from the 33 multidrug-resistant *Aeromonas* was compared using BLAST, and those with >98% similarity and the highest scores were considered for identification (Fig 5).

The phylogenetic tree based on the concatenated nucleotide sequences comprise three primary branches. Branch I consisted of only two reference strains, *A. schubertii* CECT 4240[T] and *A. enteropelogenes* CECT 4255[T] (Fig 5A). Branch II contained eight reference strains, each of which had different branches in the phylogenetic tree, making them easily distinguishable. Branch III comprised four reference strains, including *A. allosaccharophila* CECT 4199[T], *A. sobria* CECT 4245[T], *A. jandaei* CECT 4228[T], *A. veronii* CECT 4257[T], and all 33 isolates. Interestingly, six isolates, 20712 and 20705, 20503 and 20901, and 21409 and 20801, were clustered into one isolate each (Fig 5). These three groups were also assigned ST1042, ST1030, and ST1026, respectively, in the MLST analysis, with consistent results. Based on the multilocus

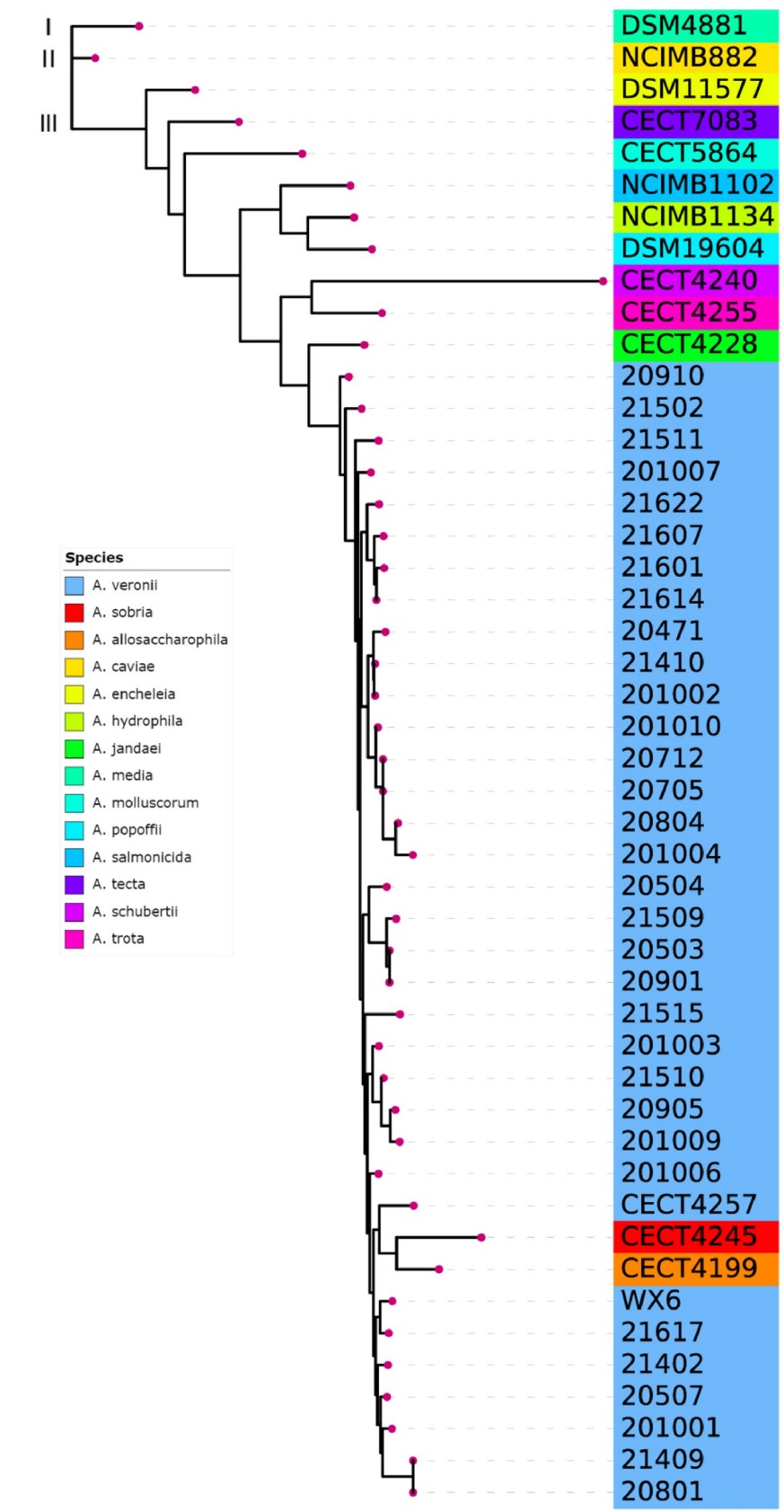

**Fig 4. PhyML phylogenetic tree based on concatenated gene sequence.** The phylogenetic tree generated by the PhyML 3.0 sofware. According to the SMS tool, the best model for the phylogenetic tree is GTR+G+I with Proportion of invariable sites = 0.583; Gamma shape parameter = 0.666; Number of substitution rate categories = 4.

sequence analysis, all 33 multidrug-resistant *Aeromonas* were identified as belonging to the *A. veronii*, which is inconsistent with the identification results based on the *16S rRNA* gene. Based on the *16S rRNA* gene, strains 20503 and 20901 were identified as *A. hydrophila*, and strain 20801 was identified as *A. media*, and strains 20905 and 21622 were identified as *A. salmonicida*.

In addition, the BLAST analysis of the six housekeeping genes displayed distinct patterns for each gene. For *gyrB* gene, seven of the eight strains were identified as *A. sobria*, unlike the other five genes, and strain 201006 was identified as *Aeromonas* sp. that could not be identified at the species level. The strains 20801 in the *gltA* group, 21409 in the *metG* group, and 21502 in the *recA* group were identified as *A. hydrophila*, and the strains 20910 and 21510 in the *recA* group were identified as *A. sobria*. The phylogenetic tree based on amino acid sequences shows branching patterns similar to the concatenated nucleotide sequences ([Fig 5B]). *A. allo-saccharophila* CECT 4199$^T$ and *A. veronii* CECT 4257$^T$ are clustered together with the 33 multidrug-resistant *Aeromonas* in Branch III. However, CECT 4257$^T$ is positioned in the middle

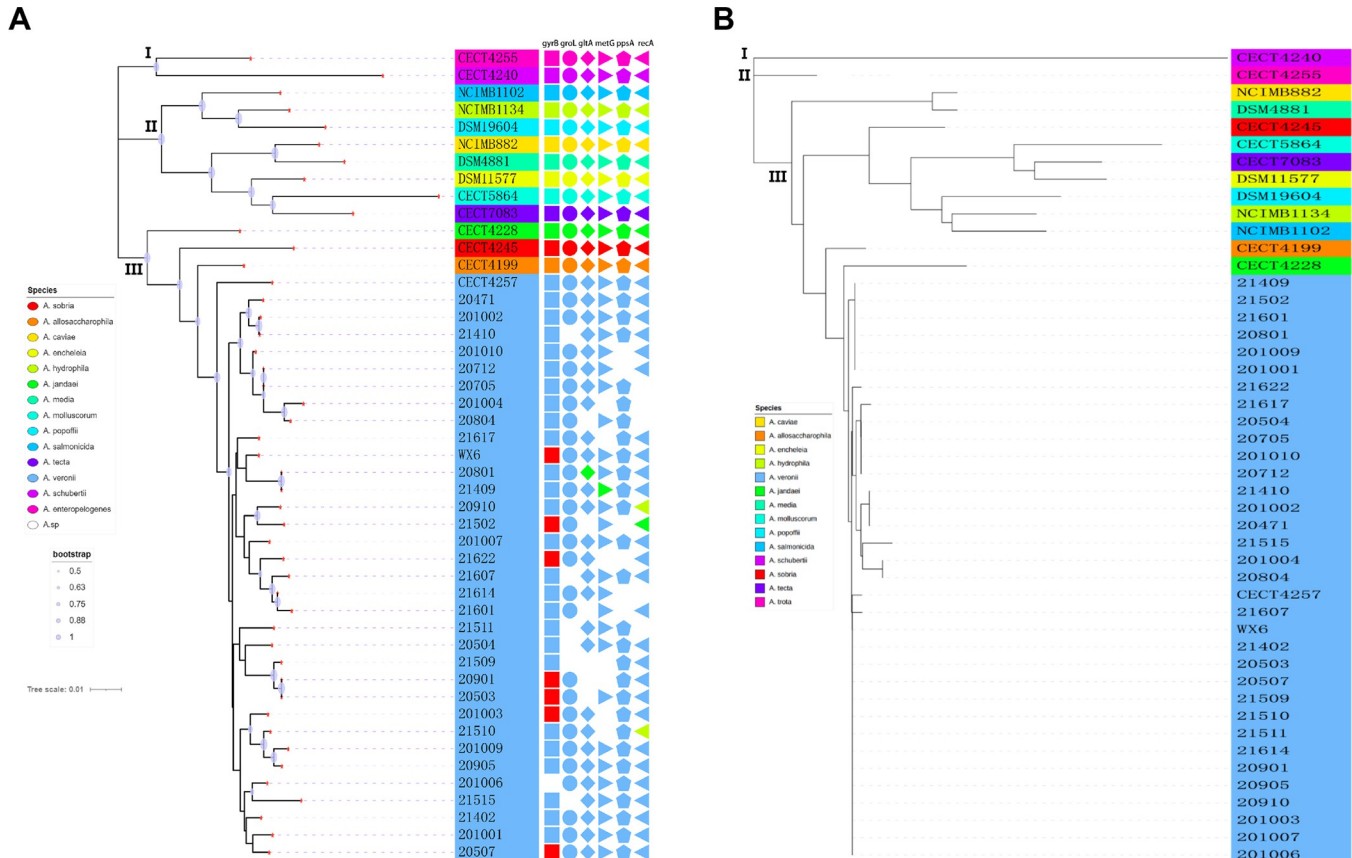

**Fig 5. Neighbor joining phylogenetic tree based on concatenated gene sequence.** Based on concatenated gene sequences, the phylogenetic tree shows the relationships among 47 strains. **(A)** A neighbor-joining phylogenetic tree based on concatenated gene nucleotide sequences shows the presence or absence of housekeeping genes for each strain. It shows identification results when each housekeeping gene is used alone. **(B)** Neighbor joining phylogenetic tree based on concatenated gene amino acid sequences.

and is closest to strain 21607. *A. sobria* CECT 4245[T] is separated from its original branch and clustered with six reference strains, including *A. molluscorum* CECT 5864[T]. *A. schubertii* CECT 4240[T] and *A. trota* CECT 4255[T] are separately classified into Branch I and II. Due to the relatively conserved nature of amino acid sequences, 14 strains of *A. veronii*, including WX6, are clustered into one branch, suggesting that nucleotide-based phylogenetic analysis may be more discriminating for sequences with high similarity or short length.

The *gltA* gene-based phylogenetic tree shows a topology similar to the concatenated sequence-based phylogenetic tree (Fig 6A). Strain CECT 4199[T] (*A. allosaccharophila*) clusters uniquely with *A. veronii* isolates in this study and reference strain CECT 4257[T]. In contrast, Branch I consists solely of strain NCIMB 882 (*A. caviae*), indicating distinct *gltA* gene characteristics for this bacterium. The *recA* gene-based phylogenetic tree presents a similar pattern, with strain 20910 and 21502 clustering with CECT 4228[T] (*A. jandaei*), consistent with strain 21502 matching as *A. jandaei* in Fig 6B. However, identification of strain 20910 as *A. hydrophila* in the BLAST result could be attributed to the similarity of the top three BLAST match scores (Fig 6B). These findings demonstrate the lower discriminatory power of BLAST comparisons compared to phylogenetic analyses and the lower discriminatory power of single-locus analyses compared to concatenated genes. The phylogenetic trees based on the single *groL* and *gyrB* genes are generally consistent with the concatenated gene phylogenetic tree, with some differences in reference strains (Fig 6C, 6D). For example, in the *gyrB*-based tree, strain CECT 4240[T] belongs to Branch I, whereas in the *groL*-based tree, it belongs to Branch II. Branch III topologies in both trees remain essentially consistent, indicating similar *groL* and *gyrB* gene sequences for strains CECT 4255[T], CECT 4245[T], CECT 4228[T], and CECT 4199[T], suggesting relative stability in these two genes during evolution. However, as shown in Fig 6A, we suggest that the phylogenetic analysis provides greater discriminatory power than *gyrB* gene-based BLAST comparisons. In the phylogenetic tree based on the *metG* gene, *A. jandaei* was included in Branch III, which includes 33 multidrug-resistant *Aeromonas*, in contrast to the concatenated tree. Unlike the *groL* based-gene tree, strain CECT 4240[T] occupies a distinctive position in the phylogenetic tree (Fig 6E). The *ppsA* tree displays a unique topology, with strain CECT 4257[T] clustered on in Branch II with *A. trota*, and *A. allosaccharophila* forming a separate branch with the tested strains (Fig 6F). Given the diverse topologies observed in the phylogenetic trees constructed based on individual loci, it is suggested that single housekeeping genes may have a limited impact on *Aeromonas* classification, with changes in concatenated genes or the entire genome serving as the primary driving factor.

## Recombination analysis

To explore recombination events in the concatenated sequences of six housekeeping genes of 47 strains, we analyzed recombination events using RDP v5.5 software with seven algorithms (RDP, GENCONV, BootScan, MaxChi, Chimaera, SiScan, and 3Seq). A total of 42 unique recombination events were detected, and 22 events were verified by three or more algorithms (S4 Table).

In addition, the maximum likelihood method of the ClonalFrameML software was used to simultaneously detect the positions of recombination regions in the isolates and clones and to reconstruct phylogenetic trees. Only ten of the 47 strains (20503, 20705, 20712, 20801, 20901, 21409, 21410, 21614, 2010021 and 201010) showed no recombination events in the concatenated genes. 49 recombination events were observed out of the 37 *Aeromonas* expected to have recombination events (Fig 7).

The average recombinant fragment length ($\delta$) among all recombination events was 621.89. The ratio of recombination rate to mutation rate ($R/\theta$) was 5.44, indicating that the difference

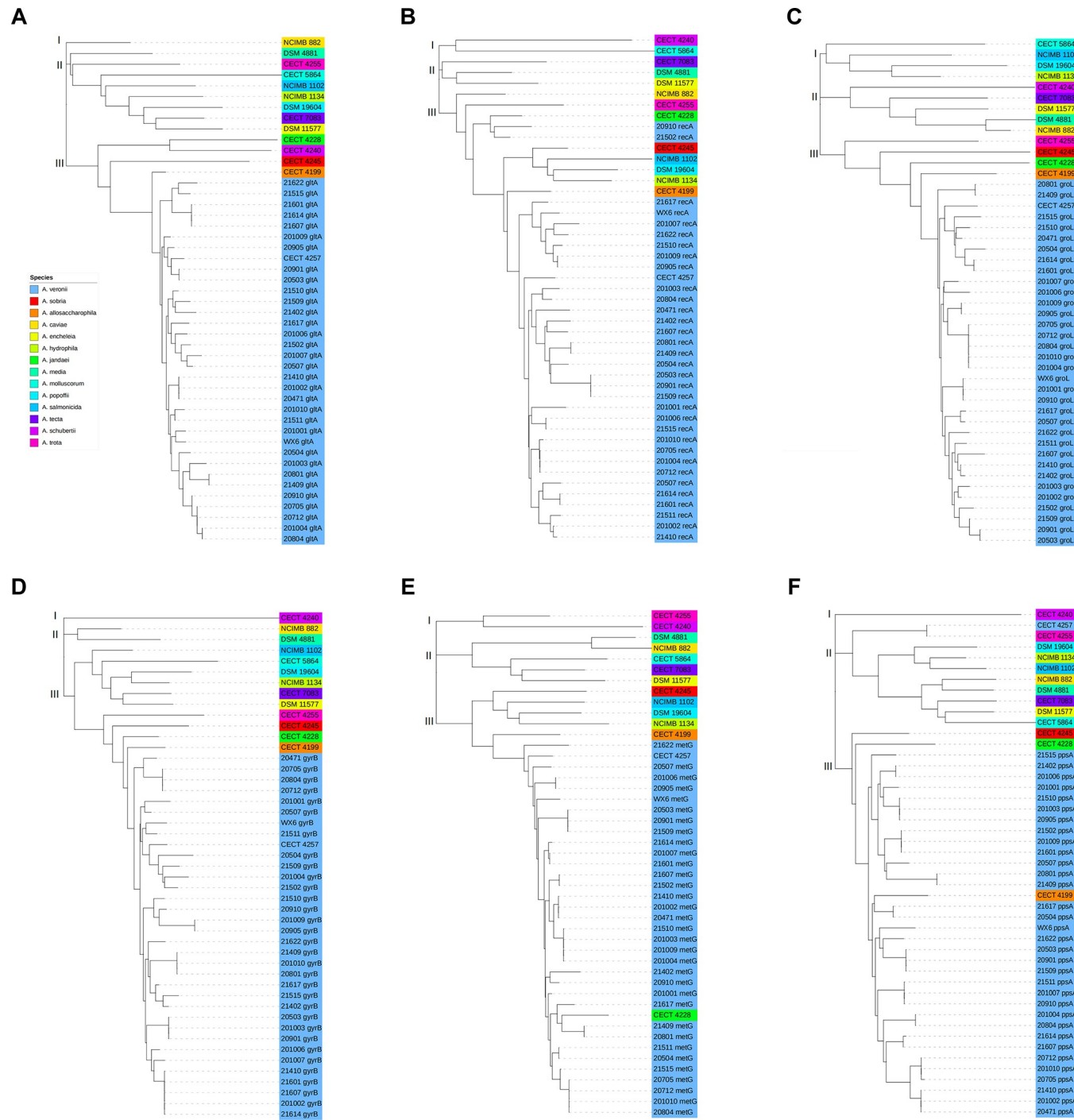

**Fig 6. Neighbor joining phylogenetic tree based on the single locus of *gltA*, *recA*, *groL*, *gyrB*, *metG* and *ppsA*.** Each phylogenetic tree represents a phylogenetic tree based on a single locus gene. (**A**) Neighbor joining phylogenetic tree based on the single locus of *gltA*. (**B**) Neighbor joining phylogenetic tree based on the single locus of *recA*. (**C**) Neighbor joining phylogenetic tree based on the single locus of *groL*. (**D**) Neighbor joining phylogenetic tree based on the single locus of *gyrB*. (**E**) Neighbor joining phylogenetic tree based on the single locus of *metG*. (**F**) Neighbor joining phylogenetic tree based on the single locus of *ppsA*.

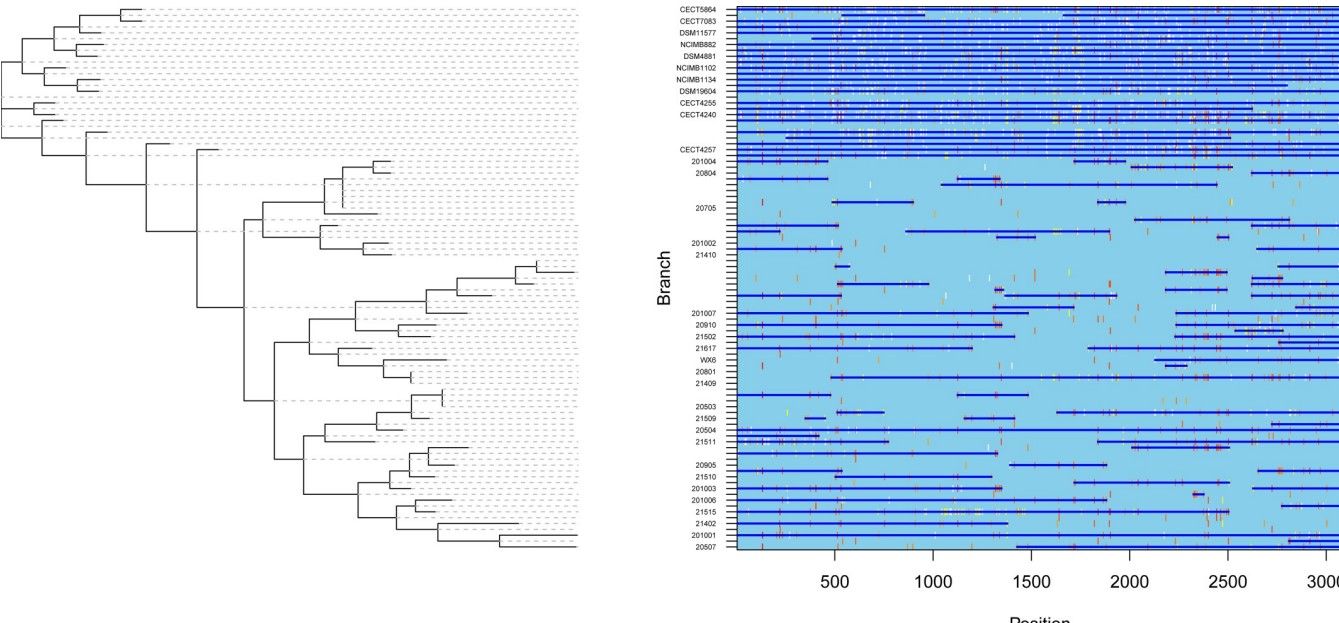

**Fig 7. Gene recombination analysis of 47 strains using ClonalFrameML.** Grey areas represent non-core regions of the 47 strains concatenated gene sequences. The concatenated gene sequences of the 47 strains were analyzed using ClonalFrameML software to detect recombination events. Recombination events were denoted by dark blue horizontal lines. Light blue positions on each branch indicated the absence of substitutions. Vertical lines of various colors from white to red indicated substitutions, where white indicated non-homoplastic substitutions and increasing shades of red indicated an increase in the degree of homoplastic substitutions.

in the probability of recombination and mutation occurring in the concatenated sequences of six housekeeping genes was insignificant. The mean divergence of imported DNA (*nu*) was 0.025, and the effect of recombination on mutation (*r/m*) was 85.69, indicating that the number of substitutions due to recombination was 85 times higher than that of mutations. Therefore, although the probability of recombination and mutation occurring in the concatenated sequences of six housekeeping genes of 47 strains was similar, this suggests that recombination played a more important role in the phylogenetic process than mutation.

## Discussion

This study identified 130 *Aeromonas* isolates collected from freshwater fish in Hebei Province, China, and confirmed the difficulty of accurate classification using existing *16S rRNA* and *gyrB* gene. Sequence analysis revealed that the genetic diversity of the *16S rRNA* gene was significantly lower than that of the *gyrB* gene, as evidenced by the Tajima's D value (D value: -1.55 < 0, *gyrB*), indicating increased population differentiation within the *gyrB* group. Furthermore, the ratio of non-synonymous to synonymous substitution rates ($K_a/K_s$) suggested that the variability of the *16S rRNA* gene was lower compared to the *gyrB* gene [61]. This indicates that the *gyrB* gene, which encodes a conserved protein of the DNA gyrase B subunit, is relatively more prone to mutations and has higher genetic diversity than the *16S rRNA* gene. Martinez et al. (2011) reported showed that the *gyrB* sequence had about six times higher the average substitution rate of the *16S rRNA* in the phylogenetic analysis of *Aeromonas* [62]. The results of our study are consistent with previous studies, showing that *gyrB* is more effective in distinguishing at the molecular biological level in *Aeromonas* spp. than *16s rRNA* [27]. Therefore, it was anticipated that the identification results using the *gyrB* gene would exhibit a significant increase in detection rate and accuracy compared to the *16S rRNA* gene, but this

expectation was not matched. This discrepancy may be attributed to factors such as the high sequence similarity between the two genes within the *Aeromonas* genus or intragenomic heterogeneity [63, 64]. In other words, the *gyrB* gene may be more suitable for the identification of *Aeromonas* spp. than the *16S rRNA* gene, but there are limitations to the method of using a single gene for a more accurate classification of *Aeromonas*.

In general, Maximum Likelihood method perform better than Neighbor-Joining method when utilized by appropriate nucleotide substitution models [65]. However, the actual construction shows that the difference between the tree construction based on the two different algorithms is relatively tiny for single gene. In the phylogenetic trees constructed for each of the two genes, *16S rRNA* gene sequences are too similar due to their conserved characteristics to allow accurate classification of branches, whereas the phylogenetic tree of the *gyrB* gene is more precise and represents a relatively accurate branching situation. Therefore, we showed that identification of *Aeromonas* using the *gyrB gene* is more reliable.

Multilocus sequence typing (MLST) analysis using six housekeeping genes provides a deeper understanding of *Aeromonas* populations and clonal complexes, improving their host preferences, habitats, and taxonomic limits [31, 66]. We identified 44 multidrug-resistant *Aeromonas* isolates from the 102 *Aeromonas* isolates and subsequently performed MLST analysis on 33 of 44 multidrug-resistant *Aeromonas*. As a result, we discovered a wealth of 30 new high abundance (ST1024-ST1053) sequence types (STs) among the 33 multidrug-resistant *Aeromonas*, which we have uploaded to the PubMLST website. eBURST analysis revealed the formation of one clonal complex (ST1027, ST1047), and the observed high nucleotide diversity suggested that most strains possessed unique sequence types, indicating a high level of genetic diversity. It is important to note that although MLST can provide valuable insights into the clonal relationships among bacterial strains, it may only capture part of the complexity of their evolutionary history [13, 67]. Finally, recombination event analysis using RDP5 and ClonalFrameML software detected 42 and 49 recombination events, respectively, with 22 results validated by four or more algorithms. Both software applications detected substantial recombination events in the nucleotide sequences of the 47 strains, with an average recombination on mutation ($r/m$) value of 85.689, underscoring the significant role of recombination events in the evolution of strains in this experiment. This indicates that the discriminatory power of a single housekeeping gene sequence alignment is insufficient, and that clone diversity increases over time due to mutation and recombination events, which can lead to incorrect typing results when only a single gene locus sequence is compared. Therefore, the use of multiple concatenated gene loci is of great importance and has higher discriminatory power.

The 102 *Aeromonas* isolates identified in this study were resistant to amoxicillin. They exhibited varying levels of resistance to several antibiotics used in the experiments, including tetracycline, cefradine, meropenem, florfenicol, and co-trimoxazole. The presence of 44 multidrug-resistant *Aeromonas*, representing for 43.14% of the 102 *Aeromonas* isolates, provides theoretical evidence for the likelihood of resistance acquisition due to irrational antibiotic use on farms rather than environmental pressures or natural selection factors [62, 68]. Analysis of the correlation between drug-resistance genotypes and phenotypes revealed that strains with specific resistance genotypes do not necessarily have resistance to all antibiotics within their respective classes. In addition, due to the limited selection of resistance genes, it is challenging to accurately match strains with specific resistance phenotypes to their corresponding genotypes. Furthermore, we also investigated the possibility that other factors, such as the sampling time, sampling location, and *Aeromonas* species could influence the resistance spectrums, but found no significant differences between the 102 *Aeromonas* isolates under different conditions. Huys et al. showed that the resistance of *Aeromonas* isolates seems to be species-independent, which is consistent with the results of this experiment [69].

In conclusion, further investigation of resistance factors at the species level, including strains, habitats, and temperature, is required to reveal the drug resistance and infection mechanisms of *Aeromonas* [70]. A deeper mechanism study requires an accurate and efficient classification method for multidrug-resistant *Aeromonas* species. Therefore, we wanted to confirm that the MLST method (multiple concatenated gene loci) was more helpful for accurate classification than the previously used single gene (*16S rRNA*, *gyrB* gene) method. Therefore, multidrug-resistant *Aeromonas* obtained from fish living in Hebei Province, China, were classified using two methods. The MLST method is suitable to contribute to the classification of multidrug-resistant *Aeromonas* species and provides theoretical information for multidrug-resistant *Aeromonas* infection prevention and antibiotic selection in the aquaculture industry in Hebei Province. The findings of this study will serve as a foundation for genetic research on the clear infection mechanism and antibiotic resistance mechanism of *Aeromonas* according to environmental conditions.

## Supporting information

**S1 Table. Inhibition zone and breakpoints.**
(XLSX)

**S2 Table. Primer information for drug resistance genes.**
(XLSX)

**S3 Table. Detection rate of tetracycline antibiotic resistance genes.**
(XLSX)

**S4 Table. Recombination events detected by RDP5.**
(XLSX)

## Acknowledgments

The authors gratefully acknowledge the Animal Ethics Committee of Hebei Normal University (Protocol Number: 198012) and Center for Bio-Medical Engineering Core Facility at Dankook University.

## Author Contributions

**Conceptualization:** Kyudong Han, Kornsorn Srikulnath, Qingyang Li, Ji-Seok Jang, Ho-Seong Lee.

**Data curation:** Zixiao Yu, Ho-Seong Lee.

**Formal analysis:** Kyudong Han, Kornsorn Srikulnath, Qingyang Li, Ji-Seok Jang.

**Funding acquisition:** Kyudong Han, Kornsorn Srikulnath, Qingyang Li, Ji-Seok Jang, Ho-Seong Lee.

**Investigation:** Zixiao Yu, Yunseok Oh, Songmi Kim, Ho-Seong Lee.

**Methodology:** Zixiao Yu.

**Project administration:** Kyudong Han, Kornsorn Srikulnath, Qingyang Li, Ji-Seok Jang, Ho-Seong Lee.

**Supervision:** Ho-Seong Lee.

**Validation:** Zixiao Yu, Yunseok Oh, Songmi Kim.

**Visualization:** Yunseok Oh, Songmi Kim.

**Writing – original draft:** Zixiao Yu, Yunseok Oh, Songmi Kim, Kyudong Han, Kornsorn Srikulnath, Qingyang Li, Ji-Seok Jang.

**Writing – review & editing:** Ho-Seong Lee.

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
