## [Decision Letter · Decision Letter 0]

4 Dec 2023

PONE-D-23-38194Antibiotic Resistance and Multilocus Sequence Typing of Aeromonas Isolated from Freshwater Fish in Hebei ProvincePLOS ONE

Dear Dr. Lee,

Thank you for submitting your manuscript to PLOS ONE. After careful consideration, we feel that it has merit but does not fully meet PLOS ONE’s publication criteria as it currently stands. Therefore, we invite you to submit a revised version of the manuscript that addresses the points raised during the review process.

We look forward to receiving your revised manuscript.

Kind regards,

Mahmoud Abdel Aziz Mabrok, PhD

Academic Editor

PLOS ONE

Journal Requirements:

4. Thank you for stating the following financial disclosure: "This research was supported by Basic Science Research Program through the National Research Foundation of Korea (NRF) funded by the Ministry of Education (NRF-RS-2023-00275307) and Basic Science Research Capacity Enhancement Project through Korea Basic Science Institute (National research Facilities and Equipment Center) grant funded by the Ministry of Education (Grant No. 2019R1A6C1010033)."

6. Please upload a new copy of Figures 6 and 7 as the detail is not clear. Please follow the link for more information: "" ext-link-type="uri" xlink:type="simple">https://blogs.plos.org/plos/2019/06/looking-good-tips-for-creating-your-plos-figures-graphics/""
"" ext-link-type="uri" xlink:type="simple">https://blogs.plos.org/plos/2019/06/looking-good-tips-for-creating-your-plos-figures-graphics/""

Reviewers' comments:

Reviewer's Responses to Questions

**Comments to the Author**

1. Is the manuscript technically sound, and do the data support the conclusions?

Reviewer #1: Yes

Reviewer #2: Yes

Reviewer #3: Yes

2. Has the statistical analysis been performed appropriately and rigorously? 

Reviewer #1: N/A

Reviewer #2: Yes

Reviewer #3: No

3. Have the authors made all data underlying the findings in their manuscript fully available?

Reviewer #1: Yes

Reviewer #2: Yes

Reviewer #3: Yes

4. Is the manuscript presented in an intelligible fashion and written in standard English?

Reviewer #1: No

Reviewer #2: Yes

Reviewer #3: Yes

5. Review Comments to the Author

Reviewer #1: Comments to authors:

-The current study is interesting; however, the authors should address the following comments to improve the quality of the manuscript:

-The manuscript should be revised for English editing and grammar mistakes.

- Please write the scientific names of bacterial pathogens and genes in the correct form all over the manuscript and the references section.

Title:

I think the work would benefit from the title that contains the main conclusion of the study (should be derived from the conclusion). Please modify the title.

Abstract:

- The abstract must illustrate the used methods and the most prevalent results (give more hints about methods and results). Besides, rephrase the aim of the work and the main conclusion of your findings.

- Add the full expression before the abbreviations.

-Introduction: (it needs to be more informative):

-Give a hint about the virulence factors and the mechanism of disease occurrence, and infecions caused by Aeromonas spp.

- The authors should illustrate the public health importance concerning the emergence of multidrug-resistant (MDR) bacterial pathogens that reflect the necessity of new potent and safe antimicrobial agents. Several studies proved the widespread MDR- bacterial pathogens;

Authors could add the following paragraph:

Multidrug resistance has been increased all over the world that is considered a public health threat. Several recent investigations reported the emergence of multidrug-resistant bacterial pathogens from different origins that increase the necessity of the proper use of antibiotics. Besides, the routine application of the antimicrobial susceptibility testing to detect the antibiotic of choice as well as the screening of the emerging MDR strains. You are advised to cite the following valuable studies:

1. PMID: 36819057

2.PMID: 36365013

3. PMID: 32235800

4. PMID: 36439817

5. PMID: 35971557

5. PMID: 35532863

6. PMID: 31170450

- Illustrate the mechanism of action different virulence factors of Aeromonads.

-Rephrase the aim of the work to be clear and better sound.

Material and methods:

- Support all methods with updated specific references.

• Add the company, city, and country of the used chemicals and reagents.

- Bacterial Strains:

Please illustrate the methods used for isolation and identification of Aeromonas spp. :

Besides, specific references should be added.

• Add the company, city, and country of the used bacterial media and reagents that were used in the biochemical identification of isolates. Also, enumerate all used biochemical reactions.

- Antimicrobial susceptibility testing:

-Please, explain in detail

•Add the names of the antimicrobial classes.

•The authors are advised to classify the tested isolates to MDR , XDR, and PDR as described by Magiorakos et al. Also, the MAR index was determined according to Krumperma.

Magiorakos AP, Srinivasan A, Carey RB, Carmeli Y, Falagas ME, Giske CG, et al. Multidrug-resistant, extensively drug-resistant and pandrug-resistant bacteria: An international expert proposal for interim standard definitions for acquired resistance. Clin Microbiol Infect. 2012; 18:268–81. doi:10.1111/j.1469-0691.2011.03570.x.

Krumperman PH. Multiple antibiotic resistance indexing of Escherichia coli to identify high-risk sources of fecal contamination of foods. Applied and Environmental Microbiology. 1983;46(1):165-70.

- The detection of virulence and antimicrobial resistance genes in the recovered isolates should be performed. Afterwards, the correlation between phenotypic and genotypic multidrug resistance should be performed.

Besides, the correlation coefficient (r) should be determined among the identified resistance genes and the tested antimicrobial agents.

-Where are the Statistical analyses? Add more details about the used software.

-Results:

- Please add a starting paragraph to the results section to briefly introduce the topic, your goals and

hypothesis and a short summary of what you did in this work.

-Add this subtitle: Phenotypic characteristics of the recovered isolates:

• Illustrate in detail the phenotypic characteristics of the recovered A. hydrophila and A. veronii isolates.

-Antimicrobial susceptibility testing

• -Illustrate in a new table the occurrence of MDR (Multidrug resistance) among the recovered isolates as the following (illustrate the names of the antimicrobial classes and different antibiotics):

No. of strains % Type of resistance

R, MDR, and XDR Phenotypic multidrug resistance

(Antimicrobial classes and different antibiotics). The antibiotic-resistance genes

-The correlation (Correlation coefficient) between phenotypic and genotypic multidrug resistance should be performed.

-Increase the resolution of figures (must be 600 dpi).

-Discussion:

- Please illustrate different mechanisms of antimicrobial resistance in Aeromonas spp.

-Conclusion

- Should be rephrased to be sounded. A real conclusion should focus on the question or claim you articulated in your study, which resolution has been the main objective of your paper?

Reviewer #2: Dear Author

Thank you for your manuscript submission. The present study is well-done and well-designed. However, a Minor Revision is needed as below:

1. The Introduction section is too long. Please do shorten it.

2. Please do add all the related references in association with the used protocols within the manuscript.

3. Please do add all the related references associated with the used primers in the present study.

4. It is recommended to add a flow chart to Materials and Methods section to show all the procedures done within this study in brief.

5. The Methodology of the manuscript is bisectional. A section is experimental and the other section is bioinformatic. Thus, it is recommended to add "Wet Lab" term for the experimental section and "Dry Lab" term for bioinformatic section.

In this regard, please do read and add the following paper to References section of the manuscript to have a fruitful Methodology section:

DNA microarray technology and bioinformatic web services. Acta Microbiol Immunol Hung. 2019 Mar 1;66(1):19-30. doi: 10.1556/030.65.2018.028. Epub 2018 Jul 16. PMID: 30010394.

6. As the Results section is interesting, it is recommended to add a schematic figure to Results section to show the most effective Results in the present study.

Reviewer #3: This study proposed that the identification selective power of Aeromonas spp. might be increased by employing the Multilocus Sequence Typing technique. However, this is an interesting work. But it has some challenges that need to be addressed before it is ready for publication.

6. PLOS authors have the option to publish the peer review history of their article (what does this mean?). If published, this will include your full peer review and any attached files.

Reviewer #1: No

Reviewer #2: **Yes: **Payam BEHZADI

Reviewer #3: No

---

## [Author Response · Author response to Decision Letter 0]

17 Jan 2024

Thank you to the Academic Editor and all Reviewers for their positive comments and interest in my Manuscript.

I have incorporated all of your suggestions into my revision. They were very helpfyl.

Upload files [Author to respond '~~~' - PLOS ONE] containing responses to Reviewer 1, 2, and 3, including the Academic Editor.

If there is a problem with the file, please contact us at any time.

---

## [Decision Letter · Decision Letter 1]

22 Jan 2024

PONE-D-23-38194R1Multilocus sequence typing and antibiotic resistance of Aeromonas isolated from freshwater fish in Hebei ProvincePLOS ONE

Dear Dr. Lee,

Thank you for submitting your manuscript to PLOS ONE. After careful consideration, we feel that it has merit but does not fully meet PLOS ONE’s publication criteria as it currently stands. Therefore, we invite you to submit a revised version of the manuscript that addresses the points raised during the review process.

In the results section, we focus exclusively on presenting our own findings without including results from other studies or any references, thus address the following comments;

**Results:**

Line 402: remove the reference (61), Page 15.

Line 413: remove the reference (32), Page 16.

Line 572: remove the reference (62), Page 21.

We look forward to receiving your revised manuscript.

Kind regards,

Mahmoud Abdel Aziz Mabrok, PhD

Academic Editor

PLOS ONE

Journal Requirements:

Reviewers' comments:

Reviewer's Responses to Questions

**Comments to the Author**

1. If the authors have adequately addressed your comments raised in a previous round of review and you feel that this manuscript is now acceptable for publication, you may indicate that here to bypass the “Comments to the Author” section, enter your conflict of interest statement in the “Confidential to Editor” section, and submit your "Accept" recommendation.

Reviewer #1: All comments have been addressed

Reviewer #2: All comments have been addressed

Reviewer #3: All comments have been addressed

2. Is the manuscript technically sound, and do the data support the conclusions?

Reviewer #1: Yes

Reviewer #2: Yes

Reviewer #3: Yes

3. Has the statistical analysis been performed appropriately and rigorously? 

Reviewer #1: Yes

Reviewer #2: Yes

Reviewer #3: Yes

4. Have the authors made all data underlying the findings in their manuscript fully available?

Reviewer #1: Yes

Reviewer #2: Yes

Reviewer #3: Yes

5. Is the manuscript presented in an intelligible fashion and written in standard English?

Reviewer #1: Yes

Reviewer #2: Yes

Reviewer #3: Yes

6. Review Comments to the Author

Reviewer #1: The authors have carried out significant changes to the manuscript. They have addressed most of the suggested corrections and comments. Really, it's an interesting study that has a significant impact. Now, the manuscript could be accepted.

Only one minor comment should be addressed in the final version: The journals names should be added in the references section.

Reviewer #2: (No Response)

Reviewer #3: Dear authors, in the Results section, we focus exclusively on presenting our own findings without including results from other studies or any references, thus address the following comments;

Results:

Line 402: remove the reference (61), Page 15.

Line 413: remove the reference (32), Page 16.

Line 572: remove the reference (62), Page 21.

7. PLOS authors have the option to publish the peer review history of their article (what does this mean?). If published, this will include your full peer review and any attached files.

Reviewer #1: No

Reviewer #2: **Yes: **Payam BEHZADI

Reviewer #3: No

---

## [Author Response · Author response to Decision Letter 1]

23 Jan 2024

Dear Reviewers

We thank the reviewers for their sincere interest and hard work.

Reviewer 3's comments in Revision Round 2 are as follows.

Response 1-3: We thank the Reviewer for pointing this out. We have corrected as Reviewer’s suggestions.

(Page 14, Line 367): [61] Huys G, Cnockaert M, Swings J. Aeromonas culicicola Pidiyar et al. 2002 is a later subjective synonym of Aeromonas veronii Hickman-Brenner et al. 1987 [J]. Systematic and applied microbiology, 2005, 28: 604-609.

(Page 14, Line 372): [32] The Clinical and Laboratory Standards Institute Subcommittee on Antimicrobial Susceptibility Testing: Background, Organization, Functions, and Processes

(Page 19, Line 525): [62] Coffey T J, Pullinger G D, Urwin R, et al. First Insights into the Evolution of Streptococcus uberis: a Multilocus Sequence Typing Scheme That Enables Investigation of Its Population Biology [J]. Applied Environmental Microbiology, 2006, 72(2): 1420-1428.

Reviewer 3's comments in Revision Round 2 are as follows.

I hope this year will be happy and healthy.

---

## [Editor Report · Decision Letter 2]

30 Jan 2024

Multilocus sequence typing and antibiotic resistance of Aeromonas isolated from freshwater fish in Hebei Province

PONE-D-23-38194R2

Dear Dr. Ho-Seong Lee

We’re pleased to inform you that your manuscript has been judged scientifically suitable for publication and will be formally accepted for publication once it meets all outstanding technical requirements.

Kind regards,

Mahmoud Abdel Aziz Mabrok, PhD

Academic Editor

PLOS ONE

---

## [Editor Report · Acceptance letter]

18 Mar 2024

PONE-D-23-38194R2 

PLOS ONE

Dear Dr. Lee, 

I'm pleased to inform you that your manuscript has been deemed suitable for publication in PLOS ONE. Congratulations! Your manuscript is now being handed over to our production team.

Kind regards, 

on behalf of

Dr. Mahmoud Abdel Aziz Mabrok 

Academic Editor

PLOS ONE